# LVTINO: LAtent Video consisTency INverse sOlver for High Definition Video Restoration

**Alessio Spagnoletti & Andrés Almansa**
Laboratoire MAP5, UMR 8145
Université Paris Cité, CNRS
F-75006 Paris, France
`alessio.spagnoletti@etu.u-paris.fr, andres.almansa@u-paris.fr`

**Marcelo Pereyra**
Heriot-Watt University, MACS & Maxwell Institute for Mathematical Sciences
EH14 4AS, Edinburgh, United Kingdom
`M.Pereyra@hw.ac.uk`

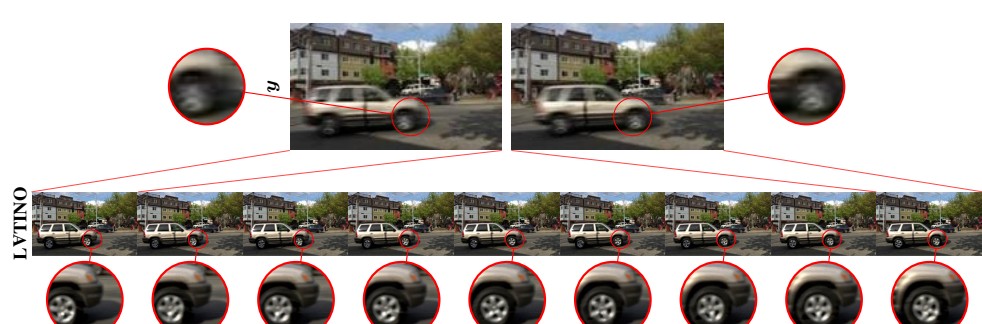

Figure 1: Results on joint spatial-temporal super-resolution by factor $\times 8$.

## ABSTRACT

Computational imaging methods increasingly rely on powerful generative diffusion models to tackle challenging image restoration tasks. In particular, state-of-the-art zero-shot image inverse solvers leverage distilled text-to-image latent diffusion models (LDMs) to achieve unprecedented accuracy and perceptual quality with high computational efficiency. However, extending these advances to high-definition video restoration remains a significant challenge, due to the need to recover fine spatial detail while capturing subtle temporal dependencies. Consequently, methods that naively apply image-based LDM priors on a frame-by-frame basis often result in temporally inconsistent reconstructions. We address this challenge by leveraging recent advances in Video Consistency Models (VCMs), which distill video latent diffusion models into fast generators that explicitly capture temporal causality. Building on this foundation, we propose LVTINO[1], the first zero-shot or plug-and-play inverse solver for high definition video restoration with priors encoded by VCMs. Our conditioning mechanism bypasses the need for automatic differentiation and achieves state-of-the-art video reconstruction quality with only a few neural function evaluations, while ensuring strong measurement consistency and smooth temporal transitions across frames. Extensive experiments on a diverse set of video inverse problems show significant perceptual improvements over current state-of-the-art methods that apply image LDMs frame by frame, establishing a new benchmark in both reconstruction fidelity and computational efficiency. The code is available on GitHub.

---

[1] LVTINO is short for LAtent Video consisTency INverse sOlver.

## 1 INTRODUCTION

We seek to recover an unknown video of interest $\boldsymbol{x} = (\boldsymbol{x}_1, \ldots, \boldsymbol{x}_T)$ from a noisy measurement

$$\boldsymbol{y} = \mathcal{A}\boldsymbol{x} + \boldsymbol{n},$$

where $\mathcal{A}$ is a linear degradation operator acting on the full video sequence, $\boldsymbol{n}$ is additive Gaussian noise with covariance $\sigma_n^2\text{Id}$, and $\boldsymbol{x}_\tau \in \mathbb{R}^n$ denotes the $\tau$th video frame.

We focus on video restoration problems that are severely ill-conditioned or ill-posed, leading to significant uncertainty about the solution. We address this difficulty by leveraging prior information about $\boldsymbol{x}$ to regularize the estimation problem and deliver meaningful solutions that are well-posed. More precisely, we adopt a Bayesian statistical approach and introduce prior information by specifying the marginal $p(\boldsymbol{x})$, so-called prior distribution, which we then combine with the likelihood function $p(\boldsymbol{y}|\boldsymbol{x}) \propto \exp\{-\|\boldsymbol{y} - \mathcal{A}\boldsymbol{x}\|_2^2 / 2\sigma_n^2\}$ by using Bayes' theorem to obtain the posterior

$$p(\boldsymbol{x}|\boldsymbol{y}) = \frac{p(\boldsymbol{y}|\boldsymbol{x})p(\boldsymbol{x})}{\int p(\boldsymbol{y}|\tilde{\boldsymbol{x}})p(\tilde{\boldsymbol{x}})\mathrm{d}\tilde{\boldsymbol{x}}}.$$

We aim to leverage a state-of-the-art generative video model as $p(\boldsymbol{x})$. In recent years, the use of deep generative models as priors in Bayesian frameworks has garnered significant attention, particularly in computational imaging, where denoising diffusion models (DMs) have emerged as powerful generative priors for solving challenging inverse problems (Song & Ermon, 2019; Song et al., 2020; Chung et al., 2022; Kawar et al., 2022; Zhu et al., 2023; Song et al., 2023a; Moufad et al., 2025).

For computational efficiency, modern DMs are often trained in the latent space of a variational autoencoder (VAE), yielding Latent Diffusion Models (LDMs), which are now the backbone of widely used large-scale priors such as Stable Diffusion (Rombach et al., 2021; Podell et al.). More recently, distilled diffusion models, and notably consistency models (CMs) (Song et al., 2023b; Luo et al., 2023a), have emerged as powerful alternatives, producing high-quality samples with only a few neural function evaluations (NFEs), in contrast to the hundreds or thousands often required by iterative DM-based methods. Several recent works have explored leveraging these models in a zero-shot, or so-called Plug & Play (PnP), manner for Bayesian computational imaging (Spagnoletti et al., 2025; Garber & Tirer, 2025; Xu et al., 2024; Li et al., 2025).

Several powerful video DMs (Ho et al., 2022; Blattmann et al., 2023b;a; Chen et al., 2023; Hong et al., 2022) and fast CMs (Wang et al., 2023; Lv et al., 2025; Zhai et al., 2024; Yin et al., 2024b) have recently been proposed, offering great potential for Bayesian video restoration. However, leveraging them remains challenging, so most current methods apply image DMs frame-by-frame and enforce temporal consistency through external constraints (Kwon & Ye, 2025a;b). In challenging settings, this strategy leads to temporal flickering and incoherent dynamics, as it fails to fully capture inter-frame dependencies. This issue could be in principle mitigated by operating directly with video DMs, but applying standard DM-guidance techniques such as DPS to video DMs requires computing gradients by backpropagation through the DM, which incurs a high memory cost (Kwon et al., 2025).

We herein present LVTINO, the first zero-shot or PnP inverse solver for Bayesian restoration of high definition videos, leveraging priors encoded by video CMs that capture fine spatial-temporal detail and causal dependencies. Moreover, by building on the recent image restoration framework of Spagnoletti et al. (2025), LVTINO provides a gradient-free inference engine that ensures strong measurement consistency and perceptual quality, while requiring few NFEs and no automatic differentiation.

## 2 BACKGROUND

We begin by revisiting the core concepts underlying DMs and LDMs, and briefly discuss their recent extension to generative modeling for video data, which we will use as priors in LVTINO.

**Diffusion Models.** (DMs) are generative models that draw samples from a distribution of interest $\pi_0(\boldsymbol{x})$ by iteratively reversing a "noising" process, which is designed to transport $\pi_0(\boldsymbol{x})$ to a standard normal distribution (Sohl-Dickstein et al., 2015; Ho et al., 2020; Song et al., 2020; Song & Ermon,

2020). In the framework of Ho et al. (2020), the noising and reverse processes are given by the SDEs:

$$d\boldsymbol{x}_t = -\frac{\beta_t}{2}\boldsymbol{x}_t dt + \sqrt{\beta_t}d\boldsymbol{w}_t, \tag{1}$$

$$d\boldsymbol{x}_t = \left[-\frac{\beta_t}{2}\boldsymbol{x}_t - \beta_t\nabla_{\boldsymbol{x}_t}\log\pi_t(\boldsymbol{x}_t)\right]dt + \sqrt{\beta_t}d\overline{\boldsymbol{w}_t}, \tag{2}$$

where $\beta_t$ is the noise schedule, and the score function $\nabla_{\boldsymbol{x}_t}\log\pi_t(\boldsymbol{x}_t)$, which encodes the target $\pi_0$, is represented by a network trained by denoising score matching on samples from $\pi_0$ (Vincent, 2011). For computational efficiency, modern DMs rely heavily on a (deterministic) probability flow representation of the backward process (2), given by the following ODE (Song et al., 2020):

$$d\boldsymbol{x}_t = \left[-\frac{\beta_t}{2}\boldsymbol{x}_t - \frac{\beta_t}{2}\nabla_{\boldsymbol{x}_t}\log\pi_t(\boldsymbol{x}_t)\right]dt. \tag{3}$$

**Latent Diffusion Models.**  LDMs dramatically increase the computational efficiency of DMs by operating in the low-dimensional latent space of an autoencoder $(\mathcal{E}, \mathcal{D})$, rather than directly in pixel space (Rombach et al., 2021). This substantially reduces compute and memory costs, enabling models like Stable Diffusion (SD) to generate large images and video (Podell et al.; Wang et al., 2025).

**Video Diffusion Models.**  Extending DMs to video is an active area of research, requiring models to capture temporal coherence and causality. Below, we highlight some key contributions to this field:

Ho et al. (2022) introduce a spatiotemporal U-Net-based DM tailored for video generation. Their architecture applies 3D convolutions to jointly process space and time, integrates spatial attention blocks for fine-grained detail, as well as temporal attention layers to capture inter-frame dependencies.

Blattmann et al. (2023b;a) propose to repurpose pre-trained LDMs to video through the incorporation of trainable temporal layers $l_i^\phi$ into a frozen U-Net backbone. The temporal layers reshape input batches into a temporally coherent sequence of frames by using a temporal self-attention mechanism.

Wang et al. (2025) introduce a state-of-the-art video foundation model built on three components: (i) *Wan-VAE*, a lightweight 3D causal variational autoencoder, inspired by Wu et al. (2024), that compresses a video $\boldsymbol{x} \in \mathbb{R}^{(1+T)\times H\times W\times 3}$ into a latent tensor $\boldsymbol{z} \in \mathbb{R}^{(1+T/4)\times H/8\times W/8\times C}$ while ensuring temporal causality; (ii) a *Diffusion Transformer (DiT)* Peebles & Xie (2022) that applies patchification, self-attention, and cross-attention to model spatio-temporal context and text conditioning; and (iii) a *text encoder* (umT5) Chung et al. (2023) for semantic conditioning. This architecture enables efficient training and scalable generation of high-resolution, temporally coherent videos.

**Consistency Models.**  Consistency Models (CMs) are single-step DM samplers derived from the probability-flow ODE (3). They rely on a so-called *consistency function* $f : (\boldsymbol{x}_t, t) \mapsto \boldsymbol{x}_\eta$ that maps any state $\boldsymbol{x}_t$ on a trajectory $\{\boldsymbol{x}_t\}_{t\in[\eta,K]}$ of (3) backwards to $\boldsymbol{x}_\eta$, for some small $\eta > 0$, ensuring $f(\boldsymbol{x}_t, t) = f(\boldsymbol{x}_{t'}, t')$ for all $t, t' \in [\eta, K]$. Two-step CMs achieve superior quality by re-noising $\boldsymbol{x}_\eta = f(\boldsymbol{x}_t, t)$ following (1) for some intermediate time $s \in (\eta, K)$, followed by $f(\boldsymbol{x}_s, s)$ to bring back $\boldsymbol{x}_s$ close to the target $\pi_0$. Multi-step CMs apply this strategy recursively in 4 to 8 steps, combining top performance with computational efficiency (Song et al., 2023b; Kim et al., 2024).

**Latent Consistency Models.**  CMs can also be trained in latent space by distilling a pre-trained LDM into a latent CM (LCM) (Luo et al., 2023a;b). A particularly effective distillation strategy is *Distribution Matching Distillation* (DMD) (Yin et al., 2023), which trains a generator $G_\theta$ to match the diffused data distribution by minimizing a KL divergence over timesteps, using a frozen teacher DM as reference. Its improved version, DMD2 (Yin et al., 2024a), adds a GAN-based loss to further enhance fidelity, and enables few-step samplers (e.g., 4 steps) by conditioning $G_\theta$ on discrete timesteps $t_i$. In practice, $G_\theta$ is often initialized from a pre-trained SDXL model (Podell et al.). We use DMD2 (Yin et al., 2024a) within our video prior, as prior distribution on individual video frames.

**Video Consistency Models.**  Recent advancements have extended CMs to video generation. Wang et al. (2023) propose VideoLCM, the first LCM framework for videos, derived by distilling a pre-trained text-to-video DM; it can generate temporally coherent videos in as few as four steps. Yin et al. (2024b) present a theoretical and practical framework to convert slow bidirectional DMs into fast

auto-regressive video generators. This conversion enables frame-by-frame causal sampling, allowing generation of very long, temporally consistent videos. Our proposed L∀TINO method incorporates the CM variant of Wan (Wang et al., 2025), distilled via DMD (Yin et al., 2023), into our video prior to effectively capture subtle spatial-temporal dependencies and long-range temporal causality.

**Zero-shot (plug & play) posteror sampling.** Zero-shot methods leverage a prior model $p(\boldsymbol{x})$ (implicit in a pretrained denoiser or generative model) and the known degradation $p(\boldsymbol{y}|\boldsymbol{x})$ to obtain an estimate of the posterior distribution $p(\boldsymbol{x}|\boldsymbol{y}) \propto p(\boldsymbol{y}|\boldsymbol{x})p(\boldsymbol{x})$. Whereas early zero-shot literature concentrates in maximum a posteriori point estimators (Venkatakrishnan et al., 2013; Monod et al., 2022), we concentrate here on producing samples from the posterior $p(\boldsymbol{x}|\boldsymbol{y})$. This has been addressed by combining prior and likelihood information in various ways, like the split Gibbs sampler (Vono et al., 2019), a discretization of the Langevin SDE (Laumont et al., 2022), a guided diffusion model (Chung et al., 2022; Zhu et al., 2023; Song et al., 2023a; Kwon & Ye, 2025a;b; Kwon et al., 2025) or a guided consistency model (Spagnoletti et al., 2025; Garber & Tirer, 2025; Xu et al., 2024; Li et al., 2025), which is the approach we pursue in this work.

LATINO (Spagnoletti et al., 2025) constructs a Markov chain approximating a Langevin diffusion $\boldsymbol{x}_s$ targeting $p(\boldsymbol{x}|\boldsymbol{y})$ by using the following splitting scheme:

$$\boldsymbol{u} = \boldsymbol{x}_k + \int_0^{\delta_k} \nabla \log p(\tilde{\boldsymbol{x}}_s)\,\mathrm{d}s + \sqrt{2}\,\mathrm{d}\boldsymbol{w}_s, \quad \tilde{\boldsymbol{x}}_0 = \boldsymbol{x}_k\,, \tag{4}$$

$$\boldsymbol{x}_{k+1} = \boldsymbol{u} + \delta_k \nabla \log p(\boldsymbol{y}|\boldsymbol{x}_{k+1})\,, \tag{5}$$

with step-size $\delta_k$. Note that the first step corresponds to an overdamped Langevin diffusion targeting the prior $p(\boldsymbol{x})$, while the second step incorporates the likelihood via an implicit Euler step.

In order to embed an LCM $(\mathcal{E}, \mathcal{D}, f_\theta)$ as prior $p(\boldsymbol{x})$, LATINO replaces (4), which is intractable, with a stochastic auto-encoder (SAE) step that applies the forward and reverse transports (1)-(3) as follows

$$\boldsymbol{z} = \sqrt{\alpha_{t_k}}\mathcal{E}(\boldsymbol{x}_k) + \sqrt{1-\alpha_{t_k}}\boldsymbol{\epsilon}\,,$$
$$\boldsymbol{u} = \mathcal{D}(f_\theta(\boldsymbol{z}, t_k))\,,$$
$$\boldsymbol{x}_{k+1} = \boldsymbol{u} + \delta_k \nabla \log p(\boldsymbol{y}|\boldsymbol{x}_{k+1})\,,$$

where we note that the SAE step preserves three fundamental properties of (4): *(i)* contraction of random iterates $\boldsymbol{x}_k$ towards the prior $p(\boldsymbol{x})$; *(ii)* $p(\boldsymbol{x})$ is the unique invariant distribution; and *(iii)* the amount of contraction is controlled via $t_k$, which plays a role analogous to the integration step-size $\delta_k$. As demonstrated in (Spagnoletti et al., 2025), LATINO exhibits high computational efficiency, requiring only a few NFEs. By leveraging a state-of-the-art SDXL LCM (Yin et al., 2024a), it achieves remarkable accuracy and perceptual quality across a range of challenging imaging tasks.

## 3  L∀TINO FOR HIGH DEFINITION VIDEO POSTERIOR SAMPLING

We are now ready to present our proposed LAtent Video consisTency INverse sOlver (L∀TINO), which approximately draws samples from the posterior distribution

$$p(\boldsymbol{x}|\boldsymbol{y}, c, \lambda) = \frac{p(\boldsymbol{y}|\boldsymbol{x})p(\boldsymbol{x}|c, \lambda)}{\int_{\mathbb{R}^n} p(\boldsymbol{y}|\boldsymbol{x})p(\boldsymbol{x}|c, \lambda)\mathrm{d}\boldsymbol{x}}\,,$$

parametrized by the data $\boldsymbol{y}$, a text prompt $c$, and a spatiotemporal regularization parameter $\lambda \in \mathbb{R}_+^3$. As mentioned previously, L∀TINO is a zero-shot Langevin posterior sampler specialised for video restoration, which jointly leverages prior information from both Video Consistency Models (VCMs) and Image Consistency Models (ICMs). In addition, L∀TINO is highly computationally efficient, requiring only a small number of NFEs and operating in a gradient-free manner, which significantly reduces memory usage and enables scalability to long video sequences.

A main novelty in L∀TINO is the use of the following product-of-experts prior for video restoration

$$p(\boldsymbol{x}|c, \lambda) \propto p_V^\eta(\boldsymbol{x}|c)p_I^{1-\eta}(\boldsymbol{x}|c)p_\phi(\boldsymbol{x}|\lambda)\,,$$

where $\eta \in (0, 1)$ is a temperature parameter and $p_V(\boldsymbol{x}|c)$, $p_I(\boldsymbol{x}|c)$, and $p_\phi(\boldsymbol{x}|\lambda)$ are as follows:

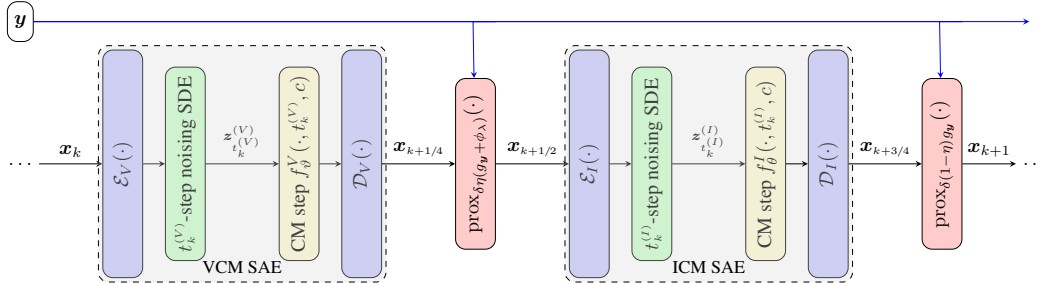

Figure 2: One step of the L∇TINO solver, a discretization of the Langevin SDE (7) which targets the posterior $p(\boldsymbol{x}|\boldsymbol{y}, c, \lambda)$, involving two stochastic autoencoding (SAE) steps and two proximal steps.

- $p_V(\boldsymbol{x}|c)$ is implicitly defined via a text-to-video LCM designed to capture subtle spatial-temporal dependencies as well as long-range temporal causality. It is specified by an encoder-decoder pair $(\mathcal{E}_V, \mathcal{D}_V)$ and consistency function $f_\vartheta^V$ operating in their latent space.

- $p_I(\boldsymbol{x}|c)$ is implicitly defined via a high-resolution text-to-image LCM, acting separately on each frame, to recover fine spatial detail and enhance perceptual quality. It is specified by an encoder-decoder pair $(\mathcal{E}_I, \mathcal{D}_I)$ and consistency function $f_\theta^I$ operating in their latent space.

- $p_\phi(\boldsymbol{x}|\lambda) \propto \exp\{-\phi_\lambda(\boldsymbol{x})\}$ where $\phi_\lambda$ is a convex regularizer promoting background stability and smooth temporal transitions across frames, with $\lambda \in \mathbb{R}_+^3$ controlling the regularity enforced. Without loss of generality, in our experiments we use the total-variation norm

$$\phi_\lambda(\boldsymbol{x}) = \mathrm{TV}_3^\lambda(\boldsymbol{x}) \triangleq \sum_{\tau,c,i,j} \sqrt{\lambda_h^2 \big(D_h \boldsymbol{x}_{\tau,c,i,j}\big)^2 + \lambda_v^2 \big(D_v \boldsymbol{x}_{\tau,c,i,j}\big)^2 + \lambda_t^2 \big(D_t \boldsymbol{x}_{\tau,c,i,j}\big)^2}.$$

where $(D_h, D_v, D_t)$ is the three-dimensional discrete gradient. Note that $\mathrm{TV}_3^\lambda$ is not smooth.

Following a PnP philosophy, $p(\boldsymbol{x}|\boldsymbol{y}, c, \lambda)$ combines an analytical likelihood function $p(\boldsymbol{y}|\boldsymbol{x})$ with a prior distribution $p(\boldsymbol{x}|c, \lambda)$ that is represented implicitly by a pre-trained machine learning model. However, unlike conventional PnP approaches that exploit a denoising operator (e.g., PnP Langevin (Laumont et al., 2022)), L∇TINO leverages the LATINO framework of Spagnoletti et al. (2025) which is specialised for embedding generative models as priors, notably distilled foundation CMs.

To draw samples from $p(\boldsymbol{x}|\boldsymbol{y}, c, \lambda)$, L∇TINO considers a Moreau-Yosida regularized overdamped Langevin diffusion, given by the SDE

$$\begin{aligned} \mathrm{d}\boldsymbol{x}_s &= \nabla \log p(\boldsymbol{y}|\boldsymbol{x}_s)\mathrm{d}s + \nabla \log p_V^\eta(\boldsymbol{x}_s|c)\mathrm{d}s + \nabla \log p_I^{(1-\eta)}(\boldsymbol{x}_s|c)\mathrm{d}s \\ &\quad + \nabla \log \tilde{p}_{\gamma\phi}(\boldsymbol{x}_s|\lambda)\mathrm{d}s + \sqrt{2}\mathrm{d}\boldsymbol{w}_s, \end{aligned} \tag{6}$$

where $\boldsymbol{w}_s$ denotes a $n$-dimensional Brownian motion and $\tilde{p}_{\gamma\phi}(\boldsymbol{x}_s|\lambda)$ is the $\gamma$-Moreau-Yosida approximation of the non-smooth factor $p_\phi(\boldsymbol{x}_s|\lambda)$, given by (Pereyra, 2016)

$$\tilde{p}_{\gamma\phi}(\boldsymbol{x}|\lambda) \propto \sup_{\boldsymbol{u} \in \mathbb{R}^n} p_\phi(\boldsymbol{u}|\lambda) \exp\{-\frac{1}{2\gamma}\|\boldsymbol{x} - \boldsymbol{u}\|_2^2\},$$

with $\gamma > 0$. As mentioned previously, $\tilde{p}_{\gamma\phi}(\boldsymbol{x}|\lambda)$ is log-concave and Lipchitz differentiable by construction because $\phi_\lambda$ is convex on $\mathbb{R}^n$ (Pereyra, 2016). The likelihood $p(\boldsymbol{y}|\boldsymbol{x}) \propto \exp\{-\|\boldsymbol{y} - \mathcal{A}\boldsymbol{x}\|_2^2/2\sigma_n^2\}$ is also log-concave and Lipchitz differentiable.

Under mild regularity assumptions on $p_V(\boldsymbol{x}|c)$ and $p_I(\boldsymbol{x}|c)$, starting from an initial condition $\boldsymbol{x}_0$, the process $\boldsymbol{x}_s$ converges to a $\gamma$-neighborhood of $p(\boldsymbol{x}|\boldsymbol{y}, c, \lambda)$ exponentially fast as $s \to \infty$ (Laumont et al., 2022). While solving (6) exactly is not possible, considering numerical approximations of $\boldsymbol{x}_s$ provides a powerful computational framework for deriving approximate samplers for $p(\boldsymbol{x}|\boldsymbol{y}, c)$.

LVTINO stems from approximating (6) by a Markov chain derived from the following recursion: given an initialization $\boldsymbol{x}_0$ and a step-size $\delta > 0$, for all $k \geq 0$,

$$\underbrace{\boldsymbol{x}_{k+1/4} = \boldsymbol{x}_k + \int_0^\delta \eta \nabla \log p_V(\tilde{\boldsymbol{x}}_s|c)\mathrm{d}s + \sqrt{2\eta}\,\mathrm{d}\boldsymbol{w}_s, \quad \tilde{\boldsymbol{x}}_0 = \boldsymbol{x}_k}_{\text{VCM prior step}}$$

$$\underbrace{\boldsymbol{x}_{k+1/2} = \boldsymbol{x}_{k+1/4} + \eta\delta\nabla \log p\left(\boldsymbol{y}|\boldsymbol{x}_{k+1/2}\right) + \eta\delta\nabla \log \tilde{p}_{\gamma\phi}\left(\boldsymbol{x}_{k+1/2}|\lambda\right)}_{\text{implicit likelihood half-step with }\phi\text{-regularization}}$$

$$\underbrace{\boldsymbol{x}_{k+3/4} = \boldsymbol{x}_{k+1/2} + \int_0^\delta (1-\eta)\nabla \log p_I(\tilde{\boldsymbol{x}}_s|c)\mathrm{d}s + \sqrt{2(1-\eta)}\,\mathrm{d}\boldsymbol{w}_s, \quad \tilde{\boldsymbol{x}}_0 = \boldsymbol{x}_{k+1/2}}_{\text{ICM prior step}}$$

$$\underbrace{\boldsymbol{x}_{k+1} = \boldsymbol{x}_{k+3/4} + (1-\eta)\delta\nabla \log p(\boldsymbol{y}|\boldsymbol{x}_{k+1}),}_{\text{implicit likelihood half-step}}$$

(7)

where we identify a splitting in which each CM prior is involved separately through exact integration (these integrals will be approximated through SAE steps), and the likelihood is involved through two implicit (backward Euler) half-steps[2].

It is worth recalling that the Langevin diffusion is a time-homogeneous process. The iterates $\boldsymbol{x}_k$ resulting from its discrete-time approximation converge to a neighborhood of $p(\boldsymbol{x}|\boldsymbol{y}, c, \lambda)$ as $k \to \infty$. Unlike DMs, these iterates do not travel backwards in time through an inhomogeneous process. Therefore, Langevin algorithms use directly the likelihood $p(\boldsymbol{y}|\boldsymbol{x}) \propto \exp\{-\|\boldsymbol{y} - \mathcal{A}\boldsymbol{x}\|_2^2/2\sigma_n^2\}$, avoiding the need to approximate the likelihood of $\boldsymbol{y}$ w.r.t. a noisy version of $\boldsymbol{x}$, as required in guided DMs like (Chung et al., 2022; Song et al., 2023a; Kwon et al., 2025).

Following Spagnoletti et al. (2025), we compute $\boldsymbol{x}_{k+1/4}$ and $\boldsymbol{x}_{k+3/4}$ approximately via SAE steps,

$$\boldsymbol{x}_{k+1/4} = \mathcal{D}^V\left(f_\vartheta^V\left(\sqrt{\alpha_{t_k^{(V)}}}\mathcal{E}_V\left(\boldsymbol{x}^{(k)}\right) + \sqrt{1 - \alpha_{t_k^{(V)}}}\boldsymbol{\epsilon}, t_k^{(V)}\right), c\right),$$

$$\boldsymbol{x}_{k+3/4} = \mathcal{D}^I\left(f_\theta^I\left(\sqrt{\alpha_{t_k^{(I)}}}\mathcal{E}_I\left(\boldsymbol{x}^{(k)}\right) + \sqrt{1 - \alpha_{t_k^{(I)}}}\boldsymbol{\epsilon}, t_k^{(I)}, c\right)\right),$$

where we recall that $(\mathcal{E}^I, \mathcal{D}^I, f^I)$ act frame-wise and that $f_\vartheta^V$ and $f_\theta^I$ have model-specific schedules.

The implicit Euler steps in (7) can be reformulated as an explicit proximal point steps as follows

$$\tilde{\boldsymbol{x}}_{k+1/2} = \underset{\boldsymbol{u}\in\mathbb{R}^n}{\arg\min}\, g_{\boldsymbol{y}}(\boldsymbol{u}) + \left(\inf_{\boldsymbol{u}'\in\mathbb{R}^n} \phi_\lambda(\boldsymbol{u}') + \tfrac{1}{2\gamma}\|\boldsymbol{u} - \boldsymbol{u}'\|_2^2\right) + \tfrac{1}{2\delta\eta}\|\tilde{\boldsymbol{x}}_{k+1/4} - \boldsymbol{u}\|_2^2,$$

$$\approx \underset{\boldsymbol{u}\in\mathbb{R}^n}{\arg\min}\, g_{\boldsymbol{y}}(\boldsymbol{u}) + \phi_\lambda(\boldsymbol{u}) + \tfrac{1}{2\delta\eta}\|\tilde{\boldsymbol{x}}_{k+1/4} - \boldsymbol{u}\|_2^2,$$

$$\tilde{\boldsymbol{x}}_{k+1} = \underset{\boldsymbol{u}\in\mathbb{R}^n}{\arg\min}\, g_{\boldsymbol{y}}(\boldsymbol{u}) + \tfrac{1}{2\delta(1-\eta)}\|\tilde{\boldsymbol{x}}_{k+3/4} - \boldsymbol{u}\|_2^2,$$

where $g_{\boldsymbol{y}} : \boldsymbol{x} \mapsto -\log p(\boldsymbol{y}|\boldsymbol{x})$ and where we have simplified the computation of $\tilde{\boldsymbol{x}}_{k+1/2}$ by assuming that $\gamma \ll \delta\eta$ (Pereyra, 2016). The optimization problems described above are strongly convex and can be efficiently approximated by using a small number of iterations of a specialized solver. In particular, to compute $\tilde{\boldsymbol{x}}_{k+1}$, we employ a few iterations of the conjugate gradient algorithm with warm-starting (Hestenes & Stiefel, 1952). For the computation of $\tilde{\boldsymbol{x}}_{k+1/2}$, we recommend using a proximal splitting optimizer (Chambolle & Pock, 2011), or a warm-started Adam optimizer (Kingma & Ba, 2014), both of which are effective in practice. Please see Appendix A.6 for more details. Refer to Algorithm 1 for more details about LVTINO, and to Figure 2 for its schematic representation.

## 4 EXPERIMENTS

**Models.** For our experiments, we implement LVTINO by using `CausVid` as VCM prior. We adopt the standard bidirectional WaN architecture, fine-tuned as a CM. The model also supports an

---

[2]While most Langevin samplers use the explicit steps and require $\delta$ to be small, the implicit steps in (7) are stable for all $\delta > 0$, allowing LVTINO to converge quickly by taking $\delta$ large, albeit with some small bias.

---

**Algorithm 1** LVTINO (LAtent Video consisTency INverse sOlver)

---

1: **given** degraded video $y$, operator $\mathcal{A}$, initialization $x_0 = \mathcal{A}^\dagger y$, video lenght $T + 1$, steps $N = 5$
2: **given** video CM $(\mathcal{E}_V, \mathcal{D}_V, f_\vartheta^V)$, image CM $(\mathcal{E}_I, \mathcal{D}_I, f_\theta^I)$, schedules $\{t_k^{(V)}, t_k^{(I)}, \delta_k, \eta, \lambda\}_{k=0}^{N-1}$, $g_y$
3: **for** $k = 0, \dots, N - 1$ **do**
4:      *# VCM prior half-step (temporal coherence)*
5:      $\epsilon_V \sim \mathcal{N}\big(0, \mathrm{Id}_{(1+T/4)\times H/8 \times W/8 \times C}\big)$
6:      $z_{t_k^{(V)}}^{(V)} \leftarrow \sqrt{\alpha_{t_k^{(V)}}}\, \mathcal{E}_V\big(x_{k-1}\big) \;+\; \sqrt{1 - \alpha_{t_k^{(V)}}}\, \epsilon_V$
7:      $\tilde{x}_{k+1/4} \leftarrow \mathcal{D}_V\big(f_\vartheta^V(z_{t_k^{(V)}}^{(V)}, t_k^{(V)})\big)$                          ▷ VCM
8:      *# First likelihood - Solved with proximal splitting or Adam iterations*
9:      $\tilde{x}_{k+1/2} \leftarrow \arg\min_{u \in \mathbb{R}^{(T+1)\times H \times W \times 3}} g_y(u) + \phi_\lambda(u) + \frac{1}{2\delta_k \eta}\|\tilde{x}_{k+1/4} - u\|_2^2$
10:      **if** $k < N$ **then**
11:          *# ICM prior half-step (per-frame detail)*
12:          $\epsilon_I \sim \mathcal{N}\big(0, \mathrm{Id}_{h/8 \times w/8 \times c}\big)$
13:          $\tilde{x}_{k+3/4} \leftarrow \mathrm{stack}_{\tau=0}^T \mathcal{D}_I\Big(f_\theta^I\big(\sqrt{\alpha_{t_k^{(I)}}}\, \mathcal{E}_I(\tilde{x}_{k+1/2, \tau}) + \sqrt{1 - \alpha_{t_k^{(I)}}}\, \epsilon_I,\; t_k^{(I)}\big)\Big)$     ▷ ICM
14:          *# Likelihood prox (2nd) - Solved with conjugate gradient iterations*
15:          $x_k \leftarrow \arg\min_{u \in \mathbb{R}^{(T+1)\times H \times W \times 3}} g_y(u) + \frac{1}{2\delta_k(1-\eta)}\|\tilde{x}_{k+3/4} - u\|_2^2$
16:      **else**
17:          *# Final iteration: skip ICM and second likelihood*
18:          $x_k \leftarrow \tilde{x}_{k+1/2}$
19:      **end if**
20: **end for**
21: **return** $x_N$

---

autoregressive configuration, which we do not utilize here, leaving the exploration of autoregressive priors for longer video restoration to future work. We use `DMD2` as ICM as Spagnoletti et al. (2025). For our experiments, we use $t_i^{(V)} \in \{757, 522, 375, 255, 125\}$ and $t_i^{(I)} \in \{374, 249, 124, 63\}$ for the VCM and ICM respectively. This results in a total of 9 NFEs, where applying the ICM across all frames counts as a single NFE. Regarding the text prompt specifying VCM and ICM, in the same spirit as Kwon & Ye (2025b), we do not perform any prompt optimization and instead use the generic prompt *"A high resolution video/image"*. Exploring prompt optimization by leveraging the maximum likelihood strategy of Spagnoletti et al. (2025) remains a key direction for future work.

**Dataset and Metrics.** We evaluate methods on 435 video clips of 25 frames each from the `Adobe240` dataset (Su et al., 2017), and 239 video clips of 25 frames each from the `GoPRO240` test dataset Nah et al. (2016). These datasets contain high-quality, high-frame-rate video sequences that we rescale to a spatial resolution of $1280 \times 768$ pixels to match our targeted resolution.

We assess reconstruction quality using peak signal-to-noise ratio (PSNR) and structural similarity index (SSIM) (Wang et al., 2004). Additionally, we evaluate two perceptual metrics: Learned Perceptual Image Patch Similarity (LPIPS) (Zhang et al., 2018), along with the recently proposed Fréchet Video Motion Distance (FVMD) (Liu et al., 2024) which is tailored for assessing motion smoothness and perceptual quality in videos.

**Inverse Problems.** We consider three linear inverse problems for high-resolution video restoration. Let $x = (x_\tau)_{\tau=0}^T \in \mathbb{R}^{(T+1)\times H \times W \times C}$ denote the unknown high-resolution video and $y = \mathcal{A}x + n$ the observed degraded video with additive Gaussian noise $n$. For fair comparisons, we consider a mild noise regime $\sigma_n = 0.001$, which addresses the noiseless case.

- **Problem A** - *Temporal SR$\times 4$ + SR$\times 4$:* here $\mathcal{A}$ first applies temporal average pooling with factor $4$ (reducing the frame rate), followed by frame-wise spatial downsampling by factor $4$, simulating a low frame rate and low resolution video. [3] Temporal upsampling to generate the missing frame is highly challenging here, as it requires prior knowledge of motion.

---

[3] Temporal SR$\times$k is also a coarse (Riemann sum) approximation of motion blur due to moving objects or camera during full continuous exposure between frames (Zhang et al., 2021).

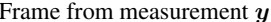
Frame from measurement $\boldsymbol{y}$

GT slice  LVTINO slice  VISION-XL slice

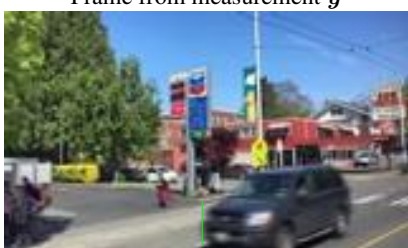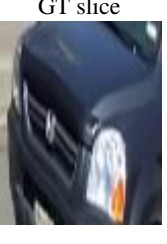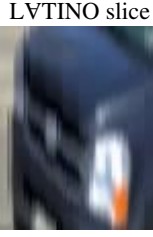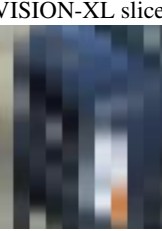

Figure 3: Comparison between slices from 81 consecutive frames for **Problem C (seq. C2)**. Slice images $(i, \tau)$ are obtained from the video tensor $(i, j, \tau)$ by fixing a column index $j$ shown in green.

- **Problem B** - *Temporal blur + SR×8:* here $\mathcal{A}$ first applies a uniform blur kernel of size 7 pixels along the temporal dimension, followed by frame-wise spatial downsampling by a factor 8, simulating a motion-blurred and low-resolution video (Kwon & Ye, 2025a;b).

- **Problem C** - *Temporal SR×8 + SR×8:* is a harder version of **Problem A**, where $\mathcal{A}$ first applies temporal average pooling with factor 8 and then a spatial downsampling by factor 8.

| Method | \multicolumn{4}{c}{Problem A: Temp. SR×4 + SR×4} | | | | \multicolumn{4}{c}{Problem B: Temp. blur + SR×8} | | | | \multicolumn{4}{c}{Problem C: Temp. SR×8 + SR×8} | | | |
|---|---|---|---|---|---|---|---|---|---|---|---|---|
| | NFE↓ | FVMD↓ | PSNR↑ | SSIM↑ | LPIPS↓ | NFE↓ | FVMD↓ | PSNR↑ | SSIM↑ | LPIPS↓ | NFE↓ | FVMD↓ | PSNR↑ | SSIM↑ | LPIPS↓ |
| LVTINO | 9 | **371.1** | **27.25** | **0.837** | **0.249** | 9 | **42.65** | 24.91 | 0.741 | **0.370** | 7 | 602.5 | 23.11 | **0.697** | **0.411** |
| VISION-XL | 8 | 1141 | 26.03 | 0.672 | 0.439 | 8 | 82.92 | **26.18** | **0.749** | 0.468 | 8 | 1604 | **23.38** | 0.652 | 0.520 |
| VIDUE | – | – | – | – | – | – | – | – | – | – | 1 | **142.5** | 21.78 | 0.624 | 0.505 |
| ADMM-TV | – | 427.6 | 18.04 | 0.767 | 0.297 | – | 128.2 | 21.18 | 0.644 | 0.452 | – | 1645 | 18.15 | 0.663 | 0.439 |

Table 1: Results on the Adobe240 dataset across the three problems. Best results are in **bold**, second best are underlined.

| Method | \multicolumn{4}{c}{Problem A: Temp. SR×4 + SR×4} | | | | \multicolumn{4}{c}{Problem B: Temp. blur + SR×8} | | | | \multicolumn{4}{c}{Problem C: Temp. SR×8 + SR×8} | | | |
|---|---|---|---|---|---|---|---|---|---|---|---|---|
| | NFE↓ | FVMD↓ | PSNR↑ | SSIM↑ | LPIPS↓ | NFE↓ | FVMD↓ | PSNR↑ | SSIM↑ | LPIPS↓ | NFE↓ | FVMD↓ | PSNR↑ | SSIM↑ | LPIPS↓ |
| LVTINO | 9 | **189.4** | 24.01 | 0.775 | **0.315** | 9 | **46.20** | 22.46 | 0.687 | **0.433** | 7 | 232.6 | 22.91 | **0.677** | **0.445** |
| VISION-XL | 8 | 282.2 | **26.06** | **0.792** | 0.326 | 8 | 52.03 | **24.05** | **0.697** | 0.486 | 8 | 995.9 | 22.67 | 0.669 | 0.474 |
| VIDUE | – | – | – | – | – | – | – | – | – | – | 1 | **84.45** | 20.66 | 0.571 | 0.548 |
| ADMM-TV | – | 265.8 | 24.32 | 0.745 | 0.406 | – | 145.9 | 20.83 | 0.618 | 0.527 | – | 969.3 | 17.70 | 0.631 | 0.527 |

Table 2: Results on the GoPro240 dataset across the three problems. Best results are in **bold**, second best are underlined.

**Computational Efficiency.** While NFEs provide a hardware-agnostic measure of complexity, practical deployment requires considering runtime and memory footprints. Table 4 reports the wall-clock time and peak GPU memory usage for restoring a 25-frame video, measured on one A100 GPU. VISION-XL, by only loading an image model, exchanges memory usage for time, as it needs to perform sequentially each frame. LVTINO offers a competitive trade-off thanks to the VCM, which scales better for longer videos. Notably, the lighter variant LVTINO-V (see Appendix A.6 for more details) achieves the fastest runtime among deep generative approaches with a moderate memory cost, as it only loads the VCM component. While VIDUE sets lower bounds on time and memory, it only solves the frame interpolation problem under uniform motion blur, and, by working natively at a lower resolution, it is not suitable for high-resolution applications.

| Method | NFE ↓ | Time (s) ↓ | Mem. (GB) ↓ |
|---|---|---|---|
| LVTINO | 9 | 132 | 35.15 |
| VISION-XL | 8 | 176 | **15.64** |
| ADMM-TV | – | 13.6 | 22.01 |
| LVTINO-V | **5** | 105 | 25.42 |
| VIDUE | 1 | 2.85 | 1.13 |

Table 3: Runtime and memory usage. Measured on a single video clip of 25 frames at $1280 \times 768$ resolution. Best results are in **bold**, second best are underlined. VIDUE is added as a light pre-trained reference.

**Results.** Experiments in Table 1 refer to **Problems A, B** and **C**, and are obtained with different numerical schemes for (7). We fix the hyperparameters per problem to better tackle the different degradations; see Table 4 in Appendix A.6 for more details and for an ablation study.

For the more challenging **Problem C**, to stabilize and warm-start LVTINO, we use the joint deblurring/interpolation network of Shang et al. (2023)[4] to produce a temporally interpolated version of $\boldsymbol{y}$,

---

[4]Which is trained on the GoPRO240 train dataset (Nah et al., 2016).

which we then upsample via bilinear spatial interpolation so that it can be used as initialization $x_0$. This warm-start allows us to reduce the number of integration steps, bringing the NFEs to 7. The same model, referred to as VIDUE, is used as a baseline comparison in Table 1 and Table 2.

We further provide a visual analysis of motion quality using fixed vertical slices of video frames, following Cohen et al. (2024), who observed that spatiotemporal slices of natural videos resemble natural images. Figure 3 and Appendix B in Figures 11a and 11b show $(i, \tau)$ slices. These reveal that even for small motions, L∀TINO more closely preserves ground truth temporal continuity.

**Qualitative and quantitative evaluation.** Figures 1, 4, 5, and 6 show the results of our algorithm compared to the measurements, ground truth and VISION-XL (see also the videos by following the links in the captions). Table 8 in Appendix B provides additional results. These results demonstrate that L∀TINO yields more detailed and temporally coherent videos than VISION-XL. The ICM prior enhances spatial detail, while the VCM prior and $TV_3^\lambda$ jointly improve temporal coherence, particularly in the challenging upsampling tasks B and C. For example, in Figure 6, L∀TINO achieves noticeably sharper results with minimal motion blur and strong temporal coherence, whereas VISION-XL shows a staircase effect with repeated frames and unresolved blur, also evident in Figure 4. In Figure 5, VISION-XL exhibits temporal flickering, which our method eliminates via the VCM and TV models. Table 1 supports these visual findings: L∀TINO achieves strong FVMD and LPIPS scores, reflecting accurate spatiotemporal dynamics and fine spatial detail.

**Other baselines.** We also report comparisons with ADMM-TV, a classical optimization-based method (we use the hyperparameters of Kwon & Ye (2025a)). We also considered comparing with VDPS (Kwon et al., 2025), however the backpropagation through Wan's DiT and Decoder at resolution $1280 \times 768$ pixels required $> 80$ Gb of VRAM, exceeding the memory capacity of GPUs available in our academic HPC facility. Since L∀TINO's conditioning mechanism does not rely on automatic differentiation, it has significantly lower memory usage. We do no include comparisons with the recently proposed zero-shot video restoration algorithm (Cao et al., 2026), which was developed concurrently with our work. Also, as our comparisons focus on zero-shot methodology, we do not report results for methods that are specifically designed and trained for a particular task, like super-resolution (Liu et al., 2025) or space-time super-resolution (Wei et al., 2026).

## 5 CONCLUSION

We introduced L∀TINO, the first VCM-based zero-shot or PnP inverse solver for Bayesian restoration of high definition videos. By combining a VCM, a frame-wise ICM and TV3 regularization, L∀TINO can recover subtle spatial temporal dynamics, as evidenced by its strong performance on challenging tasks and datasets involving both moving objects and camera shake. Moreover, L∀TINO's conditioning mechanism ensures strong measurement consistency and perceptual quality, while requiring as few as 8 NFEs and no automatic differentiation. We anticipate that upcoming advancements in distillation of VCMs will further improve the accuracy and computational efficiency of L∀TINO.

Future research will explore sequential and auto-regressive Bayesian strategies for the restoration of long videos, as well as better Langevin sampling scheme through the use of more sophisticated numerical integrators. Generative models for video tend to drift out of distribution after a few seconds (compounding errors), a problem that has only be partially addressed very recently by clever mechanisms to combine long and short-term contexts (Yesiltepe et al., 2025; Xie et al., 2025). Another promising research direction is the incorporation of automatic prompt optimization by maximum likelihood estimation, as considered in Spagnoletti et al. (2025) for image restoration tasks. Furthermore, it would be interesting to specialize L∀TINO for particular tasks through the unfolding and distillation framework of Kemajou Mbakam et al. (2025).

ACKNOWLEDGMENTS AND DISCLOSURE OF FUNDING

MP acknowledges support by UKRI Engineering and Physical Sciences Research Council (EPSRC) (EP/Z534481/1). AS and AA acknowledge support from the France 2030 research program on artificial intelligence via the PEPR PDE-AI grant (ANR-23-PEIA-0004). HPC resources provided

by GENCI-IDRIS Jean-Zay (Grant 2024-AD011014557). The authors are grateful to Murat Tekalp, Pablo Arias, Gabriele Facciolo, and Yaofang Liu for valuable discussions.

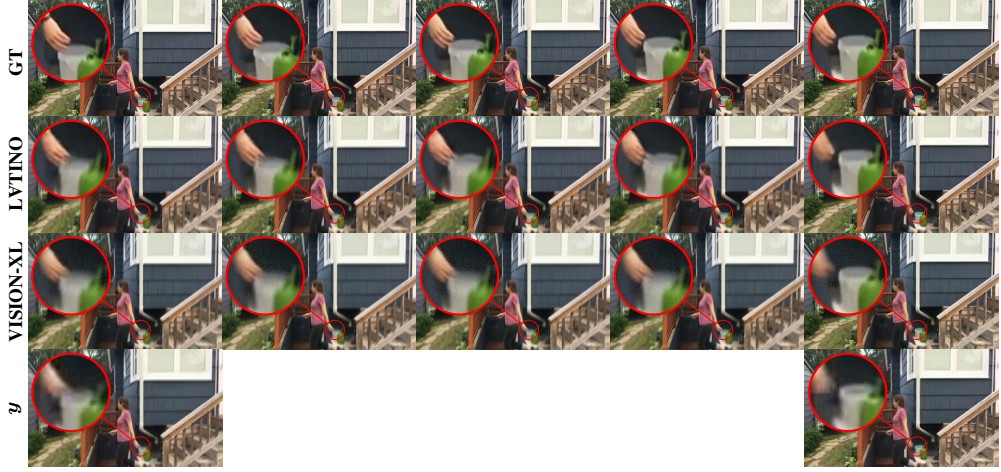

Figure 4: Visual comparison for **Problem A (seq. A1)**. The continuity of the motion is retrieved as the hand moves from right to left. See full videos: **LVTINO** and **VISION-XL**.

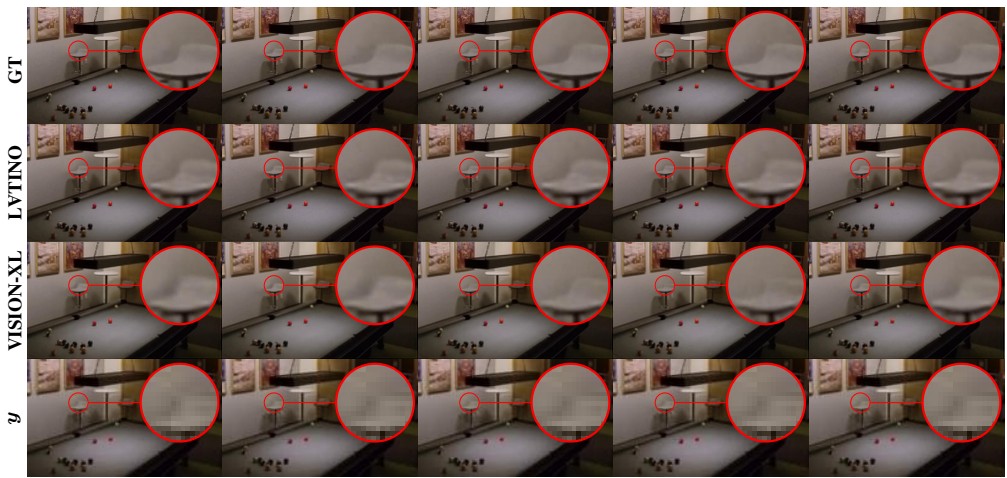

Figure 5: Visual comparison for **Problem B (seq. B2)**. The flickering problem is solved by LVTINO (see darker and lighter area behind the chair). See full videos: **LVTINO** and **VISION-XL**.

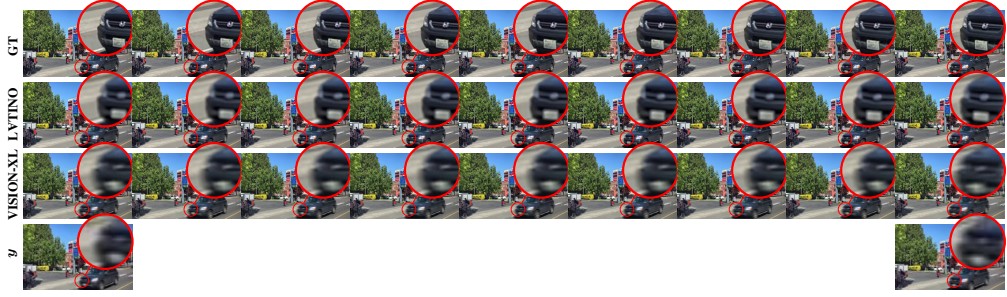

Figure 6: Visual comparison for **Problem C (seq. C2)**. The motion is retrieved by the reconstruction. See full videos (81 frames for a better direct comparison): **LVTINO** and **VISION-XL**.

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

# A  APPENDIX

## A.1  IMPLEMENTATION OF THE FORWARD OPERATORS

For all the inverse problems considered, we use the following formulation

$$\mathcal{A} = \texttt{SpatialSR} \circ \texttt{TemporalSR}.$$

For *Temporal SR×4 + Spatial SR×4*, we apply a temporal average pooling with factor $4$ (with end padding if $T$ is not divisible), followed by frame-wise spatial downsampling with factor $4$ (`DeepInv.Downsampling` Tachella et al. (2025)). The adjoint $\mathcal{A}^\top$ first applies the spatial adjoint (back-projection to HR) and then the adjoint of temporal averaging (nearest upsample by $4$ divided by $4$, with folding of the padded tail back to the last frame when $T$ is not a multiple of $4$). The same approach, but with $\times 8$, is adopted for the *Temporal SR×8 + Spatial SR×8* problem. For the *Temporal blur + Spatial SR×8* task, we use a 1D temporal uniform convolution with circular boundary conditions via FFT of window size of 7, followed by frame-wise spatial downsampling with factor 8; the adjoint corresponds to spatial back-projection and time-reversed temporal filtering via FFT.

## A.2  IMPLEMENTATION OF LIKELIHOOD PROXIMAL STEPS

We will now describe the implementation of the likelihood updates in the splitting scheme (Equation(7)) instantiated by task-specific linear operators $\mathcal{A}$ over videos $\boldsymbol{x} \in \mathbb{R}^{(T+1)\times H \times W \times 3}$. We remind that we have to solve the following problems:

$$\underset{\boldsymbol{u}\in\mathbb{R}^{(T+1)\times H \times W \times 3}}{\arg\min} g_{\boldsymbol{y}}(\boldsymbol{u}) + \phi_\lambda(\boldsymbol{u}) + \tfrac{1}{2\delta\eta}\|\tilde{\boldsymbol{x}}_{k+1/4} - \boldsymbol{u}\|_2^2, \tag{8}$$

and

$$\underset{\boldsymbol{u}\in\mathbb{R}^{(T+1)\times H \times W \times 3}}{\arg\min} g_{\boldsymbol{y}}(\boldsymbol{u}) + \tfrac{1}{2\delta(1-\eta)}\|\tilde{\boldsymbol{x}}_{k+3/4} - \boldsymbol{u}\|_2^2, \tag{9}$$

where $g_{\boldsymbol{y}}(\cdot) = \tfrac{1}{2\sigma_n^2}\|\mathcal{A}\cdot -\boldsymbol{y}\|_2^2$.

Starting from Equation (9), we notice that this is exactly the shape of the $\mathrm{prox}_{\delta(1-\eta)/2\|\mathcal{A}\cdot-\boldsymbol{y}\|_2^2}(\boldsymbol{u})$, we thus provide details about the computation of this step.

**Quadratic proximal ($\ell_2$ data term).** Given $\epsilon > 0$ (which may include $\delta, \eta$ as well as the noise variance $\sigma_n^2$), the quadratic likelihood proximal operator

$$\text{prox}_{\frac{\epsilon}{2}\|\mathcal{A}\cdot-\boldsymbol{y}\|_2^2}(\boldsymbol{u}) = \arg\min_{\boldsymbol{x}} \tfrac{\epsilon}{2}\|\mathcal{A}\boldsymbol{x} - \boldsymbol{y}\|_2^2 + \tfrac{1}{2}\|\boldsymbol{x} - \boldsymbol{u}\|_2^2$$

reduces to the normal equations

$$\left(\text{Id} + \epsilon\,\mathcal{A}^\top\mathcal{A}\right)\boldsymbol{x} \;=\; \boldsymbol{u} + \epsilon\,\mathcal{A}^\top\boldsymbol{y},$$

where Id is the identity operator. The exact solution is computationally tractable in high dimensions when $\mathcal{A}$ admits a closed-form and fast SVD (Zhang et al., 2020)[5], but to make our method applicable to general operators, we solve this linear system approximately using $\sim 10$ *Conjugate Gradient* (CG) (Hestenes & Stiefel, 1952) iterations.

CG is a Krylov-subspace method that iteratively refines an approximate solution $\boldsymbol{x}^{(k)}$ without explicitly inverting $\text{Id} + \epsilon\mathcal{A}^\top\mathcal{A}$. Starting from the initial guess $\boldsymbol{x}^{(0)} = \boldsymbol{u}$, we iteratively update:

$$\boldsymbol{r}^{(k)} = \boldsymbol{b} - \left(\text{Id} + \epsilon\mathcal{A}^\top\mathcal{A}\right)\boldsymbol{x}^{(k)}, \qquad\qquad \boldsymbol{b} := \boldsymbol{u} + \epsilon\,\mathcal{A}^\top\boldsymbol{y},$$

$$\boldsymbol{p}^{(k)} = \boldsymbol{r}^{(k)} + \beta^{(k)}\boldsymbol{p}^{(k-1)}, \qquad\qquad \beta^{(k)} := \frac{\|\boldsymbol{r}^{(k)}\|_2^2}{\|\boldsymbol{r}^{(k-1)}\|_2^2},$$

$$\alpha^{(k)} = \frac{\|\boldsymbol{r}^{(k)}\|_2^2}{\langle\boldsymbol{p}^{(k)},\,(\text{Id} + \epsilon\mathcal{A}^\top\mathcal{A})\boldsymbol{p}^{(k)}\rangle},$$

$$\boldsymbol{x}^{(k+1)} = \boldsymbol{x}^{(k)} + \alpha^{(k)}\boldsymbol{p}^{(k)}, \qquad\qquad \boldsymbol{r}^{(k+1)} = \boldsymbol{r}^{(k)} - \alpha^{(k)}\left(\text{Id} + \epsilon\mathcal{A}^\top\mathcal{A}\right)\boldsymbol{p}^{(k)}.$$

The algorithm terminates after a fixed number of iterations or once the residual norm $\|\boldsymbol{r}^{(k)}\|_2$ falls below a tolerance (e.g. $10^{-6}$). Because $\text{Id} + \epsilon\mathcal{A}^\top\mathcal{A}$ is symmetric positive definite, CG converges rapidly.

This iterative scheme is memory-efficient, requiring only matrix–vector products with $\mathcal{A}$ and $\mathcal{A}^\top$, and avoids the explicit computation of $\mathcal{A}^\top\mathcal{A}$, making it suitable for large-scale inverse problems and long video sequences.

**Spatio-temporal TV$_3$ proximal (PDHG).** For the regularised subproblem (8), we solve

$$\min_{\boldsymbol{u}} \underbrace{\tfrac{1}{2\sigma_n^2}\|\mathcal{A}\boldsymbol{u} - \boldsymbol{y}\|_2^2 + \tfrac{1}{2\delta\eta}\|\boldsymbol{u} - \tilde{\boldsymbol{x}}_{k+1/4}\|_2^2}_{f(\boldsymbol{u})} + \underbrace{\phi_\lambda(\boldsymbol{u})}_{g(D_\lambda\boldsymbol{u})}, \tag{10}$$

where

$$\phi_\lambda(\boldsymbol{u}) \;=\; \text{TV}_{3,\lambda}(\boldsymbol{u}) \;:=\; \sum_{\tau,c,i,j} \sqrt{\lambda_h^2\big(D_h\boldsymbol{u}_{\tau,c,i,j}\big)^2 + \lambda_v^2\big(D_v\boldsymbol{u}_{\tau,c,i,j}\big)^2 + \lambda_t^2\big(D_\tau\boldsymbol{u}_{\tau,c,i,j}\big)^2},$$

and $D_\lambda := \left[\lambda_h D_h,\; \lambda_v D_v,\; \lambda_\tau D_\tau\right]$, so that $g(D_\lambda\boldsymbol{u}) = \|D_\lambda\boldsymbol{u}\|_2$.

The associated subproblem in (10) is convex and can be solved using the *primal–dual hybrid gradient* (PDHG, Chambolle–Pock) algorithm Chambolle & Pock (2011). Let $\boldsymbol{p} = (p_h, p_v, p_\tau)$ denote the dual variable with three components per voxel. Given stepsizes $\rho, \sigma > 0$ such that $\rho\sigma\|D_\lambda\|^2 < 1$ and extrapolation $\theta \in [0,1]$, the iterations read:

$$\boldsymbol{p}^{k+1} = \text{prox}_{\sigma g^*}\big(\boldsymbol{p}^k + \sigma D_\lambda\bar{\boldsymbol{u}}^k\big) = \frac{\boldsymbol{p}^k + \sigma D_\lambda\bar{\boldsymbol{u}}^k}{\max\big(1,\ \|\boldsymbol{p}^k + \sigma D_\lambda\bar{\boldsymbol{u}}^k\|_2\big)} \quad \text{(projection onto unit } \ell_2 \text{ ball)},$$

$$\boldsymbol{u}^{k+1} = \text{prox}_{\rho f}\big(\boldsymbol{u}^k - \rho D_\lambda^\top\boldsymbol{p}^{k+1}\big),$$

$$\text{obtained by solving } \big(I + \rho(\mathcal{A}^\top\mathcal{A} + \tfrac{1}{\delta\eta}I)\big)\boldsymbol{u}^{k+1} = \boldsymbol{z} + \rho\Big(\mathcal{A}^\top\boldsymbol{y} + \tfrac{1}{\delta\eta}\tilde{\boldsymbol{x}}_{k+1/4}\Big),$$

$$\text{with } \boldsymbol{z} = \boldsymbol{u}^k - \rho D_\lambda^\top\boldsymbol{p}^{k+1},$$

$$\bar{\boldsymbol{u}}^{k+1} = \boldsymbol{u}^{k+1} + \theta(\boldsymbol{u}^{k+1} - \boldsymbol{u}^k).$$

---

[5]For **Problems A, B, C**, the SVD of $\mathcal{A}$ can be expressed in terms of Fourier transforms, only if convolutions are periodic, which is not always the case for the kind of spatial and temporal blur we have in our case.

Here $D_\lambda^\top \boldsymbol{p} = \lambda_h D_h^\top p_h + \lambda_v D_v^\top p_v + \lambda_\tau D_\tau^\top p_\tau$ is the weighted divergence, and the proximal step for $f(\boldsymbol{u}) = \frac{1}{2\sigma_n^2}\|\mathcal{A}\boldsymbol{u} - \boldsymbol{y}\|_2^2 + \frac{1}{2\delta\eta}\|\boldsymbol{u} - \tilde{\boldsymbol{x}}_{k+1/4}\|_2^2$ is implemented by solving the normal equations. As in our implementation $\delta\eta$ is often $\geq 10^5$, to simplify the computations we remove the regularization term $\frac{1}{2\delta\eta}\|\boldsymbol{u} - \tilde{\boldsymbol{x}}_{k+1/4}\|_2^2$. Around 10 iterations of the CG algorithm can be used to solve the normal equations, as they are warm-started with $\boldsymbol{u}^k$.

In practice, we apply Chambolle–Pock ($\sim 200$ iterations) only in the *pure temporal TV* case ($\lambda_h = \lambda_v = 0$). When spatial weights are nonzero ($\lambda_h > 0$ or $\lambda_v > 0$), we instead minimise (8) directly with ADAM (Kingma & Ba, 2014) (learning rate $10^{-3}$, 100 iterations), which proved more robust in this setting.

## A.3 THE LATINO ALGORITHM

In order to clarify the practical implementation of the splitting scheme introduced in Equation (5), we provide here the pseudo-code to implement LATINO as described in Spagnoletti et al. (2025).

---

**Algorithm 2** LATINO

---

1: **given** $\boldsymbol{x}_0 = \mathcal{A}^\dagger \boldsymbol{y}$, text prompt $c$, number of steps $N$, latent consistency model $f_\theta$, latent space decoder $\mathcal{D}$, latent space encoder $\mathcal{E}$, sequences $\{t_k, \delta_k\}_{k=0}^{N-1}$.
2: **for** $k = 0, \dots, N-1$ **do**
3:     $\boldsymbol{\epsilon} \sim \mathcal{N}(0, \mathrm{Id})$
4:     $\boldsymbol{z}_{t_k}^{(k)} \leftarrow \sqrt{\alpha_{t_k}} \mathcal{E}(\boldsymbol{x}_k) + \sqrt{1 - \alpha_{t_k}} \boldsymbol{\epsilon}$                              ▷ Encode
5:     $\boldsymbol{u}^{(k)} \leftarrow \mathcal{D}(f_\theta(\boldsymbol{z}_{t_k}^{(k)}, t_k, c))$                              ▷ Decode
6:     $\boldsymbol{x}_{k+1} \leftarrow \mathrm{prox}_{\delta_k g_y}(\boldsymbol{u}^{(k)})$                              ▷ $g_{\boldsymbol{y}} : \boldsymbol{x} \mapsto -\log p(\boldsymbol{y}|\boldsymbol{x})$
7: **end for**
8: **return** $\boldsymbol{x}_N$

---

## A.4 THE VISION-XL ALGORITHM

VISION-XL Kwon & Ye (2025b) (Video Inverse-problem Solver using latent diffusION models) is a SOTA framework for high-resolution video inverse problems, LDMs such as SDXL to restore videos from measurements affected by spatio-temporal degradations.

**Components** VISION-XL integrates three main contributions: (i) *Pseudo-batch inversion*, which initializes the sampling process from latents obtained by DDIM-inverting the measurement frames. (ii) *Pseudo-batch sampling*, which splits latent video frames and samples them in parallel using Tweedie's formula Efron (2011), reducing memory requirements to that of a single frame. (iii) *Pixel-space data-consistency updates*, where each denoised batch $\hat{\boldsymbol{x}}_t$ is refined using $l$ iterations of a quadratic proximal step

$$\bar{\boldsymbol{x}}_t = \arg \min_{\boldsymbol{x} \in \hat{\boldsymbol{x}}_t + K_l} \|\boldsymbol{y} - \mathcal{A}(\boldsymbol{x})\|_2^2,$$

typically solved via conjugate gradient (CG). This enforces alignment with the measurement before re-encoding to the latent space and re-noising for the next step.

**Overall Algorithm.** Starting from $\boldsymbol{z}_\rho = \mathrm{DDIM}^{-1}(E_\theta(\boldsymbol{y}))$ with $\rho \approx 0.3T$, VISION-XL alternates denoising in latent space and proximal data-consistency refinement in pixel space. After decoding the denoised latent batch $\hat{\boldsymbol{x}}_t = D_\theta(\hat{\boldsymbol{z}}_t)$, a low-pass filter is applied to suppress high-frequency inconsistencies before re-encoding and re-noising, yielding $\boldsymbol{z}_{t-1}$. This process is repeated until $t = 0$, as shown in Algorithm 3.

---

**Algorithm 3** VISION-XL

---

**Require:** Pretrained VAE encoder $\mathcal{E}_\theta$, decoder $\mathcal{D}_\theta$, denoiser $E_\theta^{(t)}$, measurement $\boldsymbol{x}$, forward operator $\mathcal{A}$, initial DDIM inversion step $\rho$, CG iterations $l$, low-pass filter widths $\{\sigma_t\}$, noise schedule $\{\bar{\alpha}_t\}_{t=1}^T$

1: $\boldsymbol{z}_0 \leftarrow \mathcal{E}_\theta(\boldsymbol{y})$
2: $\boldsymbol{z}_\rho \leftarrow \mathrm{DDIM}^{-1}(\boldsymbol{z}_0)$        $\triangleright$ Step 1: **Pseudo-batch inversion** (informative latent initialization)
3: **for** $t = \rho, \ldots, 2$ **do**
4:     $\hat{\boldsymbol{z}}_t \leftarrow \dfrac{\boldsymbol{z}_t - \sqrt{1 - \bar{\alpha}_t}\, E_\theta^{(t)}(\boldsymbol{z}_t)}{\sqrt{\bar{\alpha}_t}}$        $\triangleright$ Step 2: **Pseudo-batch sampling** (Tweedie's formula)
5:     $\hat{\boldsymbol{x}}_t \leftarrow \mathcal{D}_\theta(\hat{\boldsymbol{z}}_t)$
6:     $\bar{\boldsymbol{x}}_t \leftarrow \arg\min_{\boldsymbol{x} \in \hat{\boldsymbol{x}}_t + \mathcal{K}_l} \| \boldsymbol{y} - \mathcal{A}(\boldsymbol{x}) \|_2^2$   $\triangleright$ Step 3: **Data-consistency refinement** (multi-step proximal via $l$ CG steps)
7:     $\bar{\boldsymbol{x}}_t \leftarrow \bar{\boldsymbol{x}}_t * h_{\sigma_t}$   $\triangleright$ Step 4: **Scheduled low-pass filtering** (mitigate VAE error accumulation)
8:     $\bar{\boldsymbol{z}}_t \leftarrow \mathcal{E}_\theta(\bar{\boldsymbol{x}}_t)$
9:     $\boldsymbol{z}_{t-1} \leftarrow \sqrt{\bar{\alpha}_{t-1}}\bar{\boldsymbol{z}}_t + \sqrt{1 - \bar{\alpha}_{t-1}}\mathcal{E}_t$        $\triangleright$ Step 5: **Renoising** (batch-consistent noise)
10: **end for**
11: $\boldsymbol{z}_0 \leftarrow \dfrac{\boldsymbol{z}_1 - \sqrt{1 - \bar{\alpha}_1}\, E_\theta^{(1)}(\boldsymbol{z}_1)}{\sqrt{\bar{\alpha}_1}}$
12: **return** $\boldsymbol{x}_0 \leftarrow \mathcal{D}_\theta(\boldsymbol{z}_0)$

---

### A.5 CONNECTION WITH PNP-FLOW ALGORITHMS

The PnP-Flow Martin et al. (2025) algorithm designed to leverage Flow Matching image priors has some direct connections to LATINO Spagnoletti et al. (2025). For this reason, we now briefly introduce their setting and state how this idea can be extended to Video Flow models.

Let $X_0 \sim P_0$ denote a latent variable and $X_1 \sim P_1$ a data variable, with joint law $(X_0, X_1) \sim \pi$. Assume we are given a pre-trained Flow Matching model with velocity field

$$v_\theta : [0, 1] \times \mathbb{R}^d \to \mathbb{R}^d, \qquad (t, \boldsymbol{x}) \mapsto v_\theta(t, \boldsymbol{x}),$$

learned by minimizing the Conditional Flow Matching (CFM) Lipman et al. (2023) loss along the straight-line interpolation Liu et al. (2022); Benton et al. (2024)

$$X_t := e_t(X_0, X_1) := (1 - t)X_0 + tX_1, \qquad t \in [0, 1].$$

**Time-dependent denoiser from Flow Matching.** From the velocity field $v_\theta$ we define a family of time-dependent denoisers

$$D_t(\boldsymbol{x}) := \boldsymbol{x} + (1 - t)\, v_\theta(t, \boldsymbol{x}), \qquad t \in [0, 1]. \tag{11}$$

To motivate this choice, recall that for each $t \in [0, 1]$ the population minimizer $v_t^\star$ of the CFM loss satisfies

$$v_t^\star(\boldsymbol{x}) = \mathbb{E}[X_1 - X_0 \,|\, X_t = \boldsymbol{x}],$$

so that in the ideal case $v_\theta(t, \cdot) = v_t^\star(\cdot)$ one has

$$D_t(\boldsymbol{x}) = \boldsymbol{x} + (1 - t)\, v_t^\star(x) = \mathbb{E}[X_1 \,|\, X_t = \boldsymbol{x}]. \tag{12}$$

Thus $D_t$ coincides with the minimum mean-square-error (MMSE) estimator of the clean variable $X_1$ given a noisy point $X_t$ on the interpolation path. Equivalently, $D_t$ solves the regression problem

$$D_t \in \arg\min_g \mathbb{E}\big[\|X_1 - g(X_t)\|^2\big],$$

and can be interpreted as a time-indexed denoiser that projects points lying along the straight path $(X_t)_{t \in [0,1]}$ onto the target distribution $P_1$.

In particular, if the FM flow is *straight-line* in the sense that $X_t = (1 - t)X_0 + tX_1$ is realized by the associated flow ODE, then $D_t$ can perfectly recover $X_1$ from $X_t$. Under mild regularity assumptions, one can show that the mean-squared error $\mathbb{E}\big[\|D_t(X_t) - X_1\|^2\big]$ vanishes for all $t \in [0, 1]$ if and only if the learned flow forms a straight-line Flow Matching pair between $X_0$ and $X_1$.[6] This highlights the particular suitability of straight-line FM models (e.g. OT-FM Pooladian et al. (2023); Tong et al. (2024)) as building blocks for PnP priors.

---

[6]See Proposition 1 in Martin et al. (2025) for a precise statement and proof.

---

**Algorithm 4** PnP–Flow Matching

---

1: **Input:** Pre-trained Flow Matching network $v_\theta$, time sequence $(t_n)_n$ with $t_n \in [0,1]$ and $t_n \nearrow 1$, step sizes $(\gamma_n)_n$, data-fidelity $F : \mathbb{R}^d \to \mathbb{R}$, prior $P_0$ (e.g. standard Gaussian), initial iterate $\boldsymbol{x}_0 \in \mathbb{R}^d$
2: **for** $n = 0, 1, 2, \dots$ **do**
3: $\quad \boldsymbol{z}_n \leftarrow \boldsymbol{x}_n - \gamma_n \nabla F(\boldsymbol{x}_n)$ $\qquad\qquad\qquad\qquad\qquad$ ▷ gradient step on data-fidelity
4: $\quad$ Sample $\varepsilon \sim P_0$ $\qquad\qquad\qquad\qquad\qquad\qquad\qquad\qquad$ ▷ latent noise
5: $\quad \tilde{\boldsymbol{z}}_n \leftarrow (1 - t_n)\,\varepsilon + t_n\,\boldsymbol{z}_n$ $\qquad\qquad\qquad$ ▷ interpolation along the flow path
6: $\quad \boldsymbol{x}_{n+1} \leftarrow \tilde{\boldsymbol{z}}_n + (1 - t_n)\,v_\theta(t_n, \tilde{\boldsymbol{z}}_n)$ $\qquad$ ▷ PnP denoising with FM-induced denoiser $D_{t_n}$
7: **end for**
8: **Output:** Reconstruction $\boldsymbol{x}_{n+1}$

---

**PnP Flow Matching algorithm.** Martin et al. (2025) incorporates the denoisers $\{D_t\}_{t\in[0,1]}$ into a Forward–Backward Splitting (FBS) scheme for solving imaging inverse problems of the form

$$\min_{\boldsymbol{x}\in\mathbb{R}^d} F(\boldsymbol{x}) + R(\boldsymbol{x}),$$

where $F$ is a differentiable data-fidelity term (e.g. negative log-likelihood), and $R$ is an implicit prior induced by the generative model. Classical PnP-FBS Meinhardt et al. (2017); Sun et al. (2019); Hurault et al. (2022); Tan et al. (2024) replace the proximal operator of $R$ by a *time-independent* denoiser, applied directly after the gradient step on $F$.

In contrast, PnP-Flow introduces two key modifications:

1. A *time-dependent* denoiser $D_t$ as in (11), indexed by a schedule $(t_n)_n \subset [0,1]$ with $t_n \nearrow 1$.

2. An intermediate *interpolation/reprojection step* that maps the gradient iterate back onto the straight FM path before denoising.

Given an initial guess $\boldsymbol{x}_0 \in \mathbb{R}^d$, a sequence of times $(t_n)_n$ with $t_n \in [0,1]$ and $t_n \to 1$, and stepsizes $(\gamma_n)_n$, each PnP-Flow iteration at time $t_n$ proceeds as follows:

**1. Gradient step.** Move towards data consistency by a gradient descent step on $F$:

$$\boldsymbol{z}_n = \boldsymbol{x}_n - \gamma_n \nabla F(\boldsymbol{x}_n).$$

**2. Interpolation (reprojection) step.** The denoiser $D_{t_n}$ is trained to act on points distributed as $X_{t_n}$, i.e. lying on the straight-line FM path. The output $\boldsymbol{z}_n$ of the gradient step does not follow this distribution, so we "reproject" it onto the FM trajectory by drawing a latent sample $\varepsilon \sim P_0$ and forming

$$\tilde{\boldsymbol{z}}_n = (1 - t_n)\,\varepsilon + t_n\,\boldsymbol{z}_n. \tag{13}$$

Intuitively, $\tilde{\boldsymbol{z}}_n$ mimics a point at time $t_n$ on a straight path between a latent sample from $P_0$ and the current gradient iterate.

**3. PnP denoising step.** Finally, we apply the FM-induced denoiser at time $t_n$,

$$\boldsymbol{x}_{n+1} = D_{t_n}(\tilde{\boldsymbol{z}}_n) = \tilde{\boldsymbol{z}}_n + (1 - t_n)\,v_\theta(t_n, \tilde{\boldsymbol{z}}_n), \tag{14}$$

which pushes $\tilde{\boldsymbol{z}}_n$ towards the data distribution while still respecting the measurement model encoded in $F$.

The resulting discrete-time algorithm, summarized in Algorithm 4, alternates between a data-fidelity gradient step, an interpolation onto FM trajectories, and a generative PnP denoising step. The time parameter $t_n$ controls the relative weight of the prior: for small $t_n$, the denoiser has a strong effect (large factor $1 - t_n$ in (14)), while as $t_n \to 1$ the updates gradually become more likelihood-driven. Comparing Algorithm 4 to Algorithm 2, it is clear that both adapt the same core idea: data-term $\to$ add noise $\to$ denoise and repeat. Both LATINO and PnP-Flow reproject the intermediate step $\boldsymbol{x}_n$ to a point in the Flow ODE, to which is applied, in one case, the CM, and in the other, the FM denoiser. The other difference is in the type of data-fidelity term adopted; in one case, it is a proximal step as a result of an implicit Euler step, while in the other, it is a gradient one, which is equivalent to an explicit Euler step and requires many more iterations to converge due to the limitations on $\gamma_n$.

Given these similarities, it is natural to think about merging the two frameworks by leveraging few-step FMs Liu et al. (2022); Kornilov et al. (2024) in place of CMs. This would lead to a Flow-SAE that could be plugged into the LATINO algorithm and provide a different way to integrate the prior term in Equation (5). As a direct consequence, given a video FM prior, it can be deployed in place of the VCM in our Algorithm 1, and benefit from the modular framework introduced in this work, as it can be coupled with an ICM, or an image FM prior, and the TV3 term. We believe that future research may benefit from this Flow-L∇TINO formulation to improve the quality of restorations and further generalize our setting.

### A.6 ABLATION STUDY

To better understand the impact of the data-consistency updates in L∇TINO, we perform an ablation study comparing different strategies for the likelihood *proximal steps* appearing in Equation (7). Furthermore, we provide results on **Problem A** and **Problem B** obtained with a lighter version of L∇TINO that only includes the VCM prior. We call this version L∇TINO-V and we provide in Algorithm 5 its implementation.

In Table 4 we find the hyperparameters used to get Table 1 in Section 4. These values were chosen after an extensive grid search on $\lambda = (\lambda_h, \lambda_w, \lambda_\tau), \eta, \gamma$; nevertheless, other combinations also produced satisfactory results, and we want to illustrate some alternative choices in this section.

| Problem | $(\lambda_h, \lambda_v, \lambda_\tau)$ | $\eta\delta$ | $(1-\eta)\delta$ |
|---|---|---|---|
| A | $(0, 0, 2.8 \times 10^{-4})$ | $10^6$ | $10^6$ |
| B | $(0, 0, 0)$ | $3 \times 10^3$ | $1.5 \times 10^5$ |
| C | $(10^{-6}, 10^{-6}, 10^{-6})$ | $10^6$ | $10^6$ |

Table 4: Hyperparameters used in (7).

**L∇TINO: w\ and w\o TV.** As we can see from Table 4, it seems better to keep the TV prior term $\phi_\lambda$ when we solve **Problem A**, while it is better to fall back on the prox-only case (*i.e.* $\lambda = (0, 0, 0)$) when we tackle **Problem B**. We then show in Table 5 what happens in the two symmetric cases, meaning when we switch the optimal configurations of **Problem A** with those of **Problem B**. We can observe how the metrics do not change much for **Problem B**, as we are still able to beat the SOTA VISION-XL method in half of the metrics (in particular, we focus on the FVMD that tells us how temporally consistent the reconstruction is). As opposed to this, we see that we lose a lot of precision for **Problem A** in all the metrics. This can be explained by the fact that the TV prior is crucial when dealing with temporal interpolation, as it prevents the ICM from creating flickering effects.

**L∇TINO-V as a lighter alternative.** As anticipated, we also provide some results when we turn off the ICM part of the L∇TINO algorithm, meaning that we set $\eta = 1$. This solution, described in Algorithm 5, only presents choices in one data-fidelity step, which we can again tune as a TV-regularized step or as a classical prox-only step. We provide in Table 5 both cases. The values of $\lambda$ and $\delta$ are the same as Table 4, meaning that the TV case will follow the **Problem A** row and the prox case the **Problem B** row. We see how this lighter version can still beat VISION-XL in almost all metrics with only 5 NFEs. In particular, since we no longer have the ICM, the TV prior loses its importance, and the prox case emerges as the best option. L∇TINO-V is capable of getting highly temporally coherent reconstructions, as shown by the low FVMD values, only losing to L∇TINO, especially in LPIPS, as its single frame quality suffers from the limitations of the VCM. We believe that further research could fill the gap between L∇TINO and L∇TINO-V, developing new SOTA methods that solely use VCMs, without the need for its image counterpart, to increase spatial quality.

| | | Temp. SR×4 + SR×4 | | | | Temp. blur + SR×8 | | | |
|---|---|---|---|---|---|---|---|---|---|
| Method (Data-Consistency Config) | NFE↓ | FVMD↓ | PSNR↑ | SSIM↑ | LPIPS↓ | FVMD↓ | PSNR↑ | SSIM↑ | LPIPS↓ |
| **L∇TINO-V** (prox) | **5** | 425.2 | 25.00 | 0.811 | 0.270 | **31.70** | 23.80 | 0.737 | 0.375 |
| **L∇TINO (ICM: prox, VCM: prox)** | 9 | 607.5 | 22.59 | 0.614 | 0.475 | 42.65 | 24.91 | 0.741 | **0.370** |
| **L∇TINO-V (TV)** | **5** | 503.3 | 24.44 | 0.776 | 0.338 | 578.0 | 22.01 | 0.684 | 0.441 |
| **L∇TINO (ICM: prox, VCM: TV)** | 9 | **371.1** | **27.25** | **0.837** | **0.249** | 51.52 | 23.18 | 0.725 | 0.418 |
| **VISION-XL** | 8 | 1141 | 26.03 | 0.672 | 0.439 | 82.92 | **26.18** | **0.749** | 0.468 |
| **ADMM-TV** | – | 427.6 | 18.04 | 0.767 | 0.297 | 128.2 | 21.18 | 0.644 | 0.452 |

Table 5: **Ablation study on data-consistency schemes.** Left block: results for *temporal SR×4 + SR×4*, **Problem A**. Right block: results for *temporal blur + SR×8*, **Problem B**.

---

**Algorithm 5** L∀TINO-V

---

1: **given** degraded video $\boldsymbol{y}$, operator $\mathcal{A}$, initialization $\boldsymbol{x}_0 = \mathcal{A}^\dagger \boldsymbol{y}$, video lenght $T+1$, steps $N=5$
2: **given** video CM $(\mathcal{E}_V, \mathcal{D}_V, f_\vartheta^V)$, schedules $\{t_k, \delta_k, \lambda\}_{k=0}^{N-1}$, $g_{\boldsymbol{y}}$
3: **for** $k = 0, \ldots, N-1$ **do**
4: $\quad \boldsymbol{\epsilon} \sim \mathcal{N}(0, \mathrm{Id}_{(1+T/4)\times H/8 \times W/8 \times C})$
5: $\quad \boldsymbol{z}_{t_k}^{(k)} \leftarrow \sqrt{\alpha_{t_k}}\, \mathcal{E}_V(\boldsymbol{x}_k) + \sqrt{1 - \alpha_{t_k}}\, \boldsymbol{\epsilon}$ $\qquad\qquad\qquad$ ▷ encode & diffuse to $t_k$
6: $\quad \tilde{\boldsymbol{x}}_{k+1/2} \leftarrow \mathcal{D}_V(f_\vartheta^V(\boldsymbol{z}_{t_k}^{(k)}, t_k))$ $\qquad\qquad\qquad\qquad$ ▷ VCM prior contraction
7: $\quad \boldsymbol{x}_{k+1} \leftarrow \arg\min_{\boldsymbol{u} \in \mathbb{R}^{(T+1)\times H \times W \times 3}}\, g_y(\boldsymbol{u}) + \phi_\lambda(\boldsymbol{u}) + \frac{1}{2\delta_k}\|\tilde{\boldsymbol{x}}_{k+1/2} - \boldsymbol{u}\|_2^2$ ▷ data-consistency
$\qquad$ Solved with a few CG iters; TV-in-time can be used here.
8: **end for**
9: **return** $\boldsymbol{x}_N$

---

### A.7 ADDITIONAL EXPERIMENTS AND ANALYSES

**Comparisons to other baselines.** To provide a more comprehensive evaluation, we extend our comparison to include non-zero-shot methods, such as VIDUE Shang et al. (2023), which is explicitly trained for joint motion-blur removal and frame interpolation. This makes it a highly relevant baseline for the combined blur and interpolation tasks of **Problem C**, whereas standard Video Frame Interpolation (VFI) methods often fail to address motion blur. We indeed specifically compare VIDUE against the recent BiM-VFI Seo et al. (2025). As shown in Figure 7, because BiM-VFI is trained specifically for interpolation, it fails to remove the degradation caused by motion blur. In contrast, VIDUE addresses the joint problem more effectively. As VIDUE does not perform super-resolution, we apply bicubic upsampling (×8) to its output for fair comparison to L∀TINO. The results are shown in Table 1 and Table 2.

We also acknowledge that DiffIR2VR Yeh et al. (2025) is a relevant competitor to VISION-XL, and thus to L∀TINO. However, the specific `Stable Diffusion v2.1` checkpoint required to reproduce their method is no longer publicly available, which prevents a fair comparison.

**BiM-VFI** $\qquad\qquad\qquad\qquad\qquad\qquad\qquad\qquad\qquad\qquad$ **VIDUE**

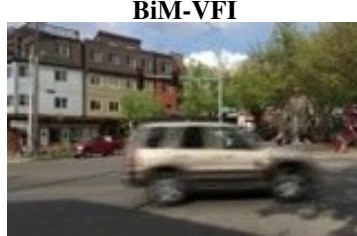 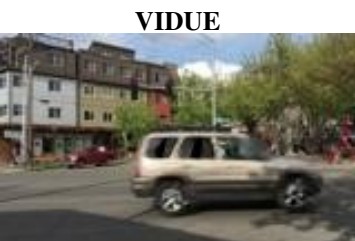

Figure 7: Visual comparison on **Problem C**. Left: BiM-VFI preserves blur artifacts. Right: VIDUE removes some motion blur.

**Noisier cases.** We now show results computed on the Adobe240 dataset for a higher noise scenario with $\sigma_{\boldsymbol{y}} = 0.01$. As expected, the optimization step in VISION-XL fails to properly restore the video sequences in this case, as VISION-XL is not conceived to deal with noisy measurements, yielding `NaN` values. In contrast, L∀TINO and ADMM-TV handle this case without difficulty. Their results are reported in Table 6, together with VIDUE for **Problem C**.

| Method | Problem A: Temp. SR×4 + SR×4 | | | | Problem B: Temp. blur + SR×8 | | | | Problem C: Temp. SR×8 + SR×8 | | | |
|---|---|---|---|---|---|---|---|---|---|---|---|---|
| | NFE↓ | FVMD↓ | PSNR↑ | SSIM↑ | LPIPS↓ | NFE↓ | FVMD↓ | PSNR↑ | SSIM↑ | LPIPS↓ | NFE↓ | FVMD↓ | PSNR↑ | SSIM↑ | LPIPS↓ |
| L∀TINO | 9 | **256.5** | **24.95** | **0.782** | **0.331** | 9 | **62.6** | **21.91** | **0.671** | **0.448** | 7 | 310.6 | **23.42** | **0.688** | **0.428** |
| VIDUE | – | – | – | – | – | – | – | – | – | – | 1 | **121.5** | 21.33 | 0.603 | 0.511 |
| ADMM-TV | – | 424.1 | 17.85 | 0.758 | 0.373 | – | 145.5 | 21.35 | 0.646 | 0.471 | – | 1665 | 18.12 | 0.652 | 0.475 |

Table 6: Results on the Adobe240 dataset with noise $\sigma_{\boldsymbol{y}} = 0.01$ across the three problems. Best results are in **bold**, second best are underlined.

**Non-linear Inverse Problems.** Although for presentation clarify we present L∀TINO in the context of linear inverse problems, L∀TINO can be applied to non-linear problems too. The main requirement

is the ability to evaluate the proximal operator of the log-likelihood, which is feasible for many non-linear degradations, as already shown in Spagnoletti et al. (2025).

To demonstrate this, we consider a non-linear degradation: Additive Gaussian noise ($\sigma = 0.01$) followed by JPEG compression (quality=10) applied to each frame independently. Figure 8 shows frames extracted from the reconstruction results. LVTINO successfully recovers high-frequency details and suppresses compression artifacts, confirming its applicability to non-linear inverse problems.

| GT | Measurement $y$ | LVTINO |
|---|---|---|

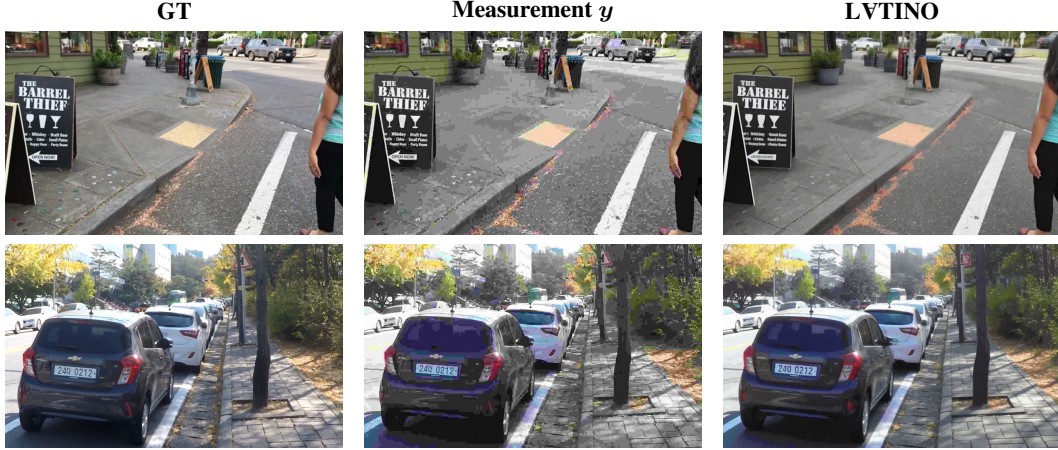

Figure 8: Results on a non-linear inverse problem (Gaussian Noise + JPEG compression). Top row: Example from Adobe240. Bottom row: Example from GoPRO240. LVTINO effectively removes blocking artifacts and noise in both cases.

**Hyperparameter Sensitivity.** We analyze the stability of LVTINO with respect to the step size $\delta$ and the regularization weight $\lambda$. Figure 9 plots PSNR and LPIPS metrics on a representative sequence from the challenging **Problem C**. We observe that performance remains stable across a reasonable range of values (e.g., $\delta \in [2 \cdot 10^4, 2 \cdot 10^5]$). This indicates that the parameters reported in Table 4 are not brittle, and $\epsilon$-good hyperparameters can be found without exhaustive fine-tuning.

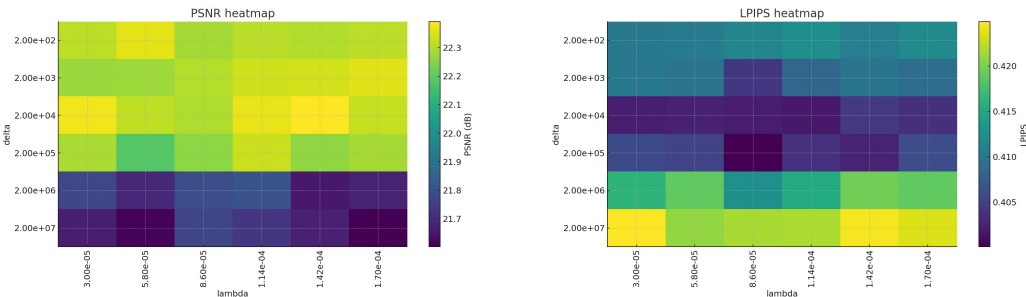

Figure 9: Sensitivity analysis for **Problem C**. The method shows robust performance across a wide range of step sizes ($\delta$) and regularization weights ($\lambda$).

In a similar way, it is also possible to analyse the parameter $\eta$, which controls the balance between the VCM and the ICM in our theoretical framework (see equation (6)). It must be translated into practice by choosing the corresponding evaluation times $t_V$ and $t_I$. In particular, when $\eta$ increases, the $t_V$ is larger, and the ICM is evaluated at a smaller $t_I$. Because pretrained Consistency Models are only accurate on a restricted subset of timesteps, this severely limits how finely we can tune $\eta$ in practice.

To approximate different effective values of $\eta$, we therefore perform an ablation in which we vary the possible video timesteps $t_V$ and image timesteps $t_I$ within the valid finetuned ranges of the two backbones. Operationally, we choose among the subsets:

- **VCM (video model):** $([757, 522, 375, 255, 125])$

- **ICM (image model):** ([749, 624, 499, 374, 249, 124, 63])

and pairing them to simulate "larger $\eta$" (larger VCM steps + smaller ICM steps) and "smaller $\eta$" (smaller VCM steps + larger ICM steps).

For clarity, Table 7 shows some configurations evaluated to provide a comparison for **Problem B** (Temporal SR×4 + SR×4). Each configuration is evaluated on the same sample sequence:

| Experiment | $t_V$ (video) | $t_I$ (image) | PSNR ↑ | SSIM ↑ | LPIPS ↓ |
|---|---|---|---|---|---|
| **EXP 1** | [375, 255, 125] | [499, 374] | 22.90 | 0.752 | 0.296 |
| **EXP 2** | [757, 522, 375] | [249, 124] | 22.56 | 0.714 | 0.308 |
| **EXP 3** | [522, 375, 255] | [374, 249] | 22.80 | 0.762 | 0.290 |
| **EXP 4** | [522, 255, 125] | [749, 624] | 22.36 | 0.720 | 0.317 |
| **BASELINE** | [757, 522, 375, 255, 125] | [374, 249, 124, 63] | 23.96 | 0.770 | 0.272 |
| **VISION-XL** | — | — | 24.36 | 0.667 | 0.488 |

Table 7: Ablation study on scheduling strategies ($t_V$ and $t_I$) for **Problem B**. EXP 1-4 represent varying balances of $\eta$, while BASELINE represents the configuration used in the main paper.

For comparison, the values used for the experiments shown in the other tables are: $t_V \in [757, 522, 375, 255, 125]$ and $t_I \in [374, 249, 124, 63]$. We notice how, even with fewer steps and varying the configurations, the metrics remain stable.

**Error Map Analysis.** To better visualize the nature of the residuals, we provide $L_2$ error maps in Figure 10 for **Problem C** on an example sequence. Comparing LVTINO against VISION-XL and ADMM-TV, we observe that our method yields lower residuals, particularly around motion boundaries and fine structural details where competing methods exhibit larger errors due to unresolved blur or temporal inconsistencies.

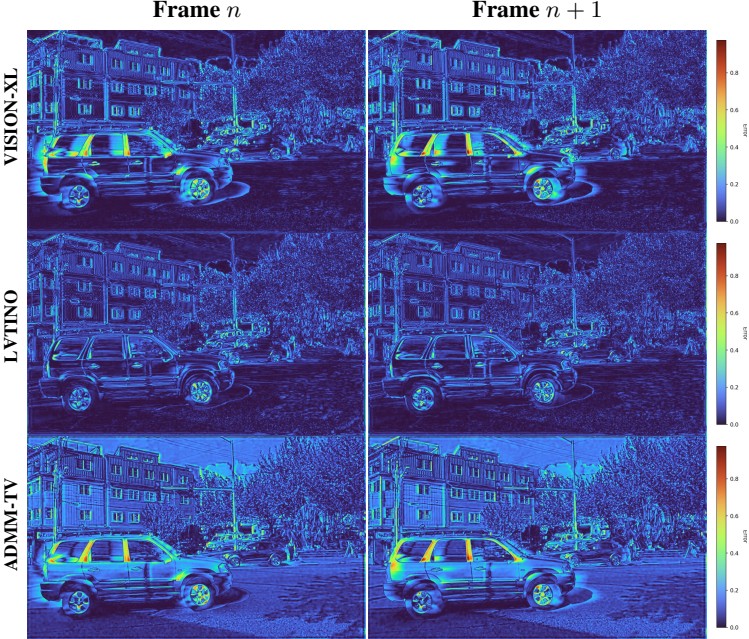

Figure 10: $L_2$ error maps between reconstructions and Ground Truth. LVTINO (middle row) demonstrates lower error magnitude compared to VISION-XL and ADMM-TV, particularly in dynamic regions.

## B    ADDITIONAL EXAMPLES

We provide in Table 8 qualitative video comparisons for **Problem A**, **Problem B**, and **Problem C**. Each triplet corresponds to the Ground Truth (GT), the observed degraded input ($y$), and the restored sequence. For **Problem C**, we provide longer sequences (81 frames) to better appreciate the results.

|  | **GT** | $y$ | **LVTINO** | **VISION-XL** |
|---|---|---|---|---|
| **Problem A (seq. A1)** | link | link | link | link |
| **Problem B (seq. B1)** | link | link | link | link |
| **Problem B (seq. B2)** | link | link | link | link |
| **Problem C (seq. C1)** | link | link | link | link |
| **Problem C (seq. C2)** | link | link | link | link |

Table 8: Results of our method compared to those obtained by VISION-XL, ground truth, and measurements (input sequence). Click the links to see the videos.

Additional examples are shown in Figures 12,13,14,15,16. We also include additional sliced images in Figures 11a and 11b.

**Frame from measurement** $y$      **GT slice**      **L∇TINO slice**      **VISION-XL slice**

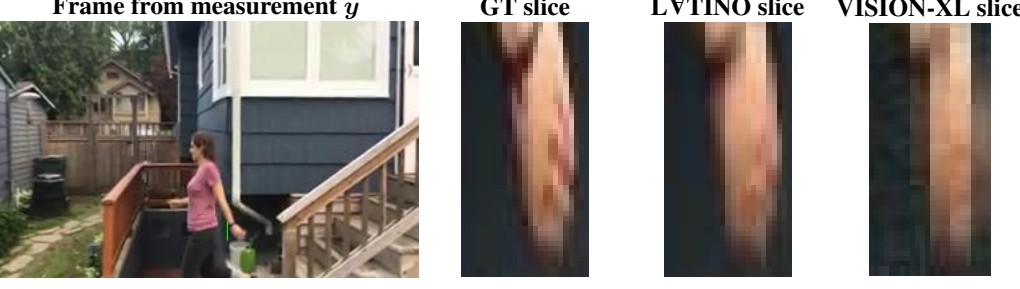

(a) Comparison between slices from 25 consecutive frames. **Problem A (seq. A1)**

**Frame from measurement** $y$      **GT slice**      **L∇TINO slice**      **VISION-XL slice**

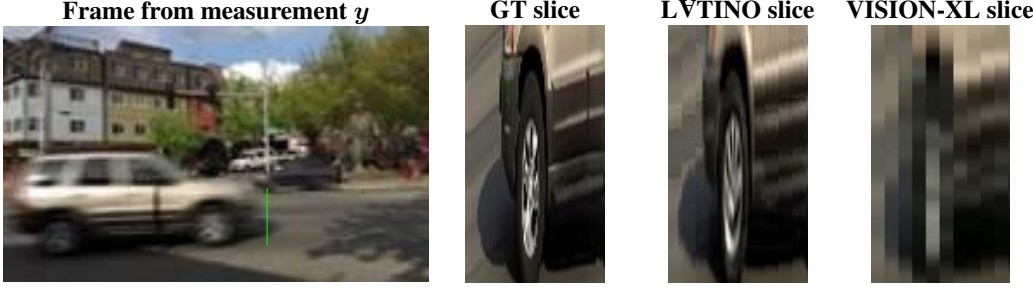

(b) Comparison between slices from 81 consecutive frames. **Problem C (seq. C1)**

Figure 11: Slice comparisons across two sequences. In green, the sliced column. Slice images are obtained from the three-dimensional video tensor $(i, j, \tau)$ by fixing a column index j. This leads to a 2D tensor with indices $(i, \tau)$ that is represented as an image, where the i index represents the row and the t index represents the column.

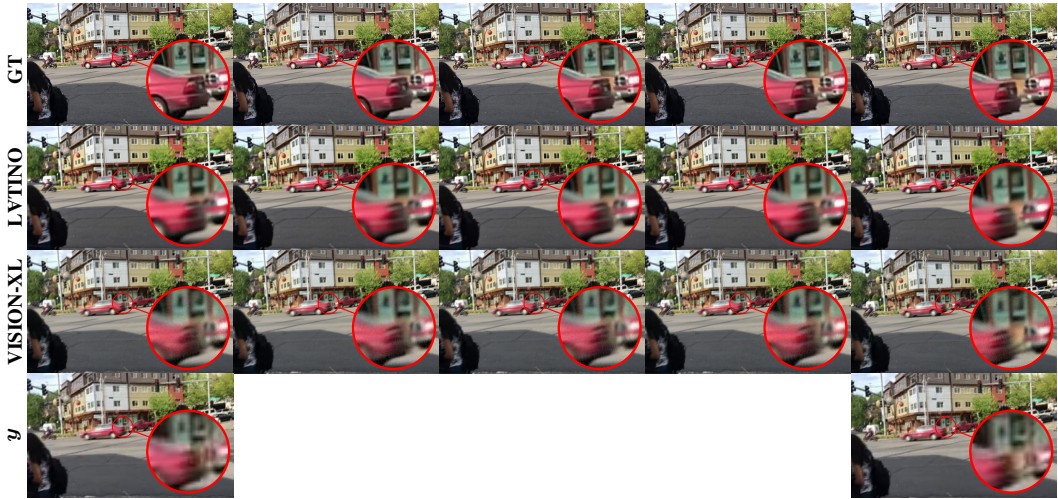

Figure 12: Visual comparison for **Problem A**.

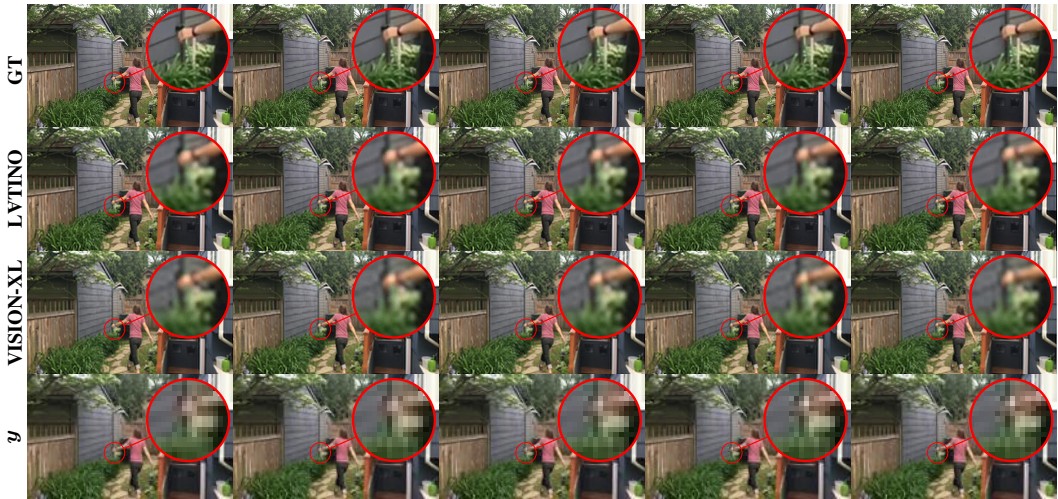

Figure 13: Visual comparison for **Problem B**.

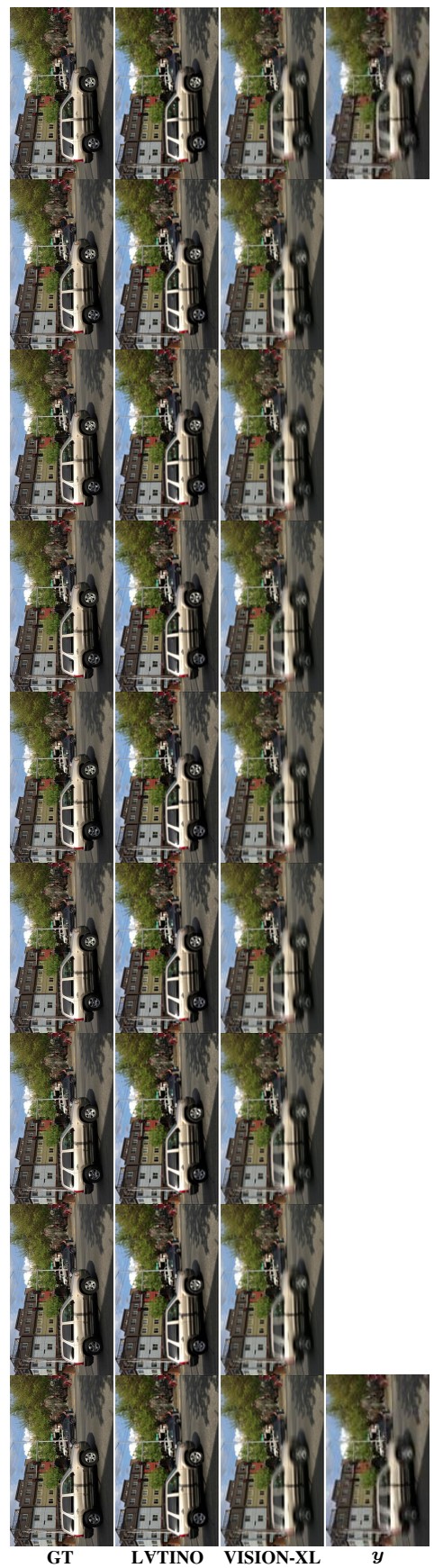

Figure 14: Visual comparison for **Problem C**.

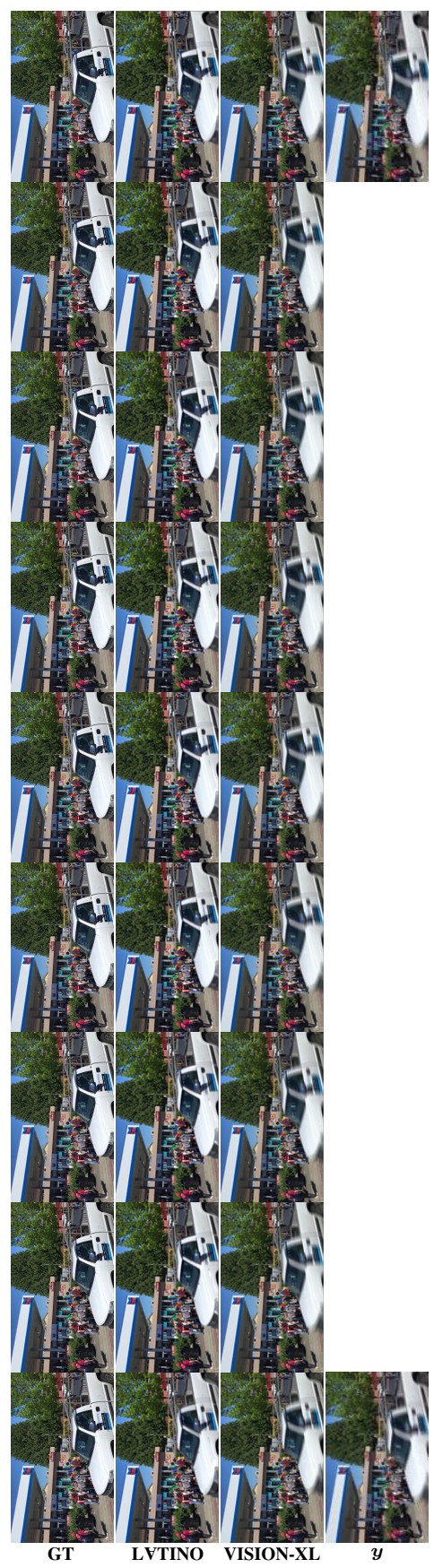

Figure 15: Visual comparison for **Problem C**.

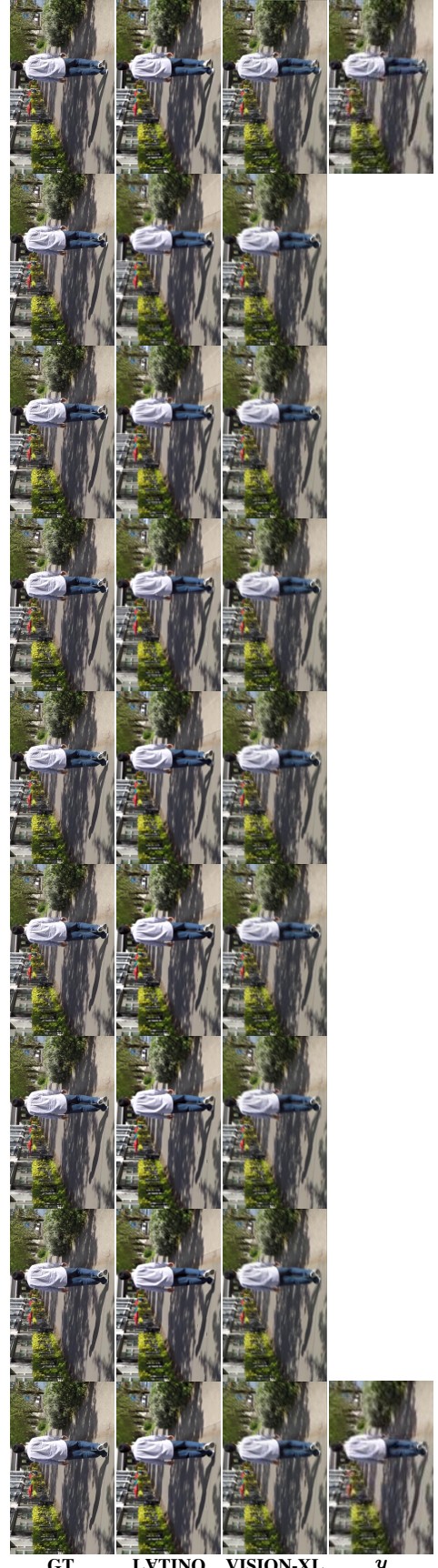

Figure 16: Visual comparison for **Problem C**.

GT      LVTINO    VISION-XL    *y*

