# OpenReview forum: "LVTINO: LAtent Video consisTency INverse sOlver for High Definition Video Restoration"
_ICLR.cc/2026/Conference — ICLR 2026 Poster_

### Official Review · Reviewer_acCb · 2025-10-24

**Soundness:** 3
**Presentation:** 1
**Contribution:** 3
**Rating:** 6
**Confidence:** 3

**Summary:**

This paper produces LVTINO, a video reconstruction model, based on Latino. It "handcrafts" a prior, using state of the art generative models. This special mixture of experts prior has the consequence that it is capturing the video as a whole object, instead of operating on a frame to frame basis.  Furthermore, relying only on "denoising" operations, this algorithm does not need to backprop through any ODE or even network evaluation. The resulting algorithm is applied to video datasets, and compared against two other models.

**Strengths:**

I find the idea very innovative. Indeed, training a full usual generative model, on a video space is prohibitively expensive. So defining a mixture of experts prior (based on generative models), one for "smoothness", one for the temporal relations, and one for frame consistency. The evaluation is good, it is nice that the authors performed a full grid search for finding suitable hyperparameters, and also showed that others work too, solidifying the trust in the method. Further, I commend the authors for providing many examples and an explanation of the latino algorithm. Furthermore, that it is gradient free makes that it is super cheap as opposed to gradient based diffusion restoration algorithms.

**Weaknesses:**

The downside is that this paper is very densely written. I dont find the intro to diffusion/consistency models helpful i would rather want to paper to introduce the latino framework and then delve into the video model. I dont think the long equation 7 is necessary and the spelling out of the Euler steps, too. Rather focus on more explanation in a simpler setting to make it easier to digest.

 There is a plethora of hyperparameters. While the grid search is nice, and good that there area other vals I would like to see that this method is not overengineered. If put on a compute (tuning budget), how does it compare to the other models?

Does the efficiacy of the approach depend a lot on the chosen consistency/distillation strategy?

Why is the hyperparameter between the video regularizer and the image one? I would rather see a tuning tradeoff between the smoothing regularizer and the image one?

**Questions:**

see weaknesses. Overall I really appreciate the novelty and lightweightness of the idea :) my main issues are making sure the algorithm is robust and understandable.

---

> ### Author Response · Authors · 2025-11-20
>
> We thank the reviewer for the very thoughtful and constructive feedback, and for the positive assessment of the originality and practicality of our approach.
>
> ---
>
> ## Weakness 1 – Densely written / introduction and Equation (7)
>
> We fully agree that readability is crucial, especially for a method that builds on several existing ideas. In the revision we will:
>
> * **Reorganize the introduction and background.**
>   We will reduce the details in the general overview of diffusion/consistency models, and introduce **LATINO** giving more details on its components.
>
> * **Shorten and move technical derivations.**
>   The full derivation around the current Equation (7) and the Euler steps will be **moved to the appendix**, where interested readers can find the detailed connection to the underlying Langevin formulation.
>   In the main text we will:
>
>   * Replace the long formula by a **shorter, high-level update rule**, and
>   * Emphasize the **intuitive roles** of each step (VCM update for temporal coherence, ICM update for spatial detail, TV step for smoothing) rather than the algebra.
>
> ---
>
> ## Weakness 2 – Many hyperparameters / grid search and “engineering cost”
>
> We agree that the number of hyperparameters is an important practical concern. Our intention was to **separate conceptual design** (VCM + ICM + TV and the alternating scheme) from **practical tuning** (step size, TV weight, mixture weight, etc.), but we recognize that this may not have been sufficiently emphasized.
>
> In the revision, we will clarify that:
>
> * We will add plots (as mentioned in another response) showing how PSNR and LPIPS change when varying **δ** (the step size) and **λ** (the TV coefficient):
>
> https://gcdnb.pbrd.co/images/UVzQ1A9QNgI7.png?o=1
>
>   These graphs are relatively flat, indicating that **near-optimal performance is obtained for a broad range of settings**. This supports the view that LVTINO is not overly fragile despite having several controls. The values in Table 2 have been tuned on more extensive tests averaged on the Adobe240 dataset, but for a generic video a simple and small grid search can be enough to find $\varepsilon$-optimal hyperparameters.
>
> We will also highlight more that LVTINO benefits from a transparent, modular structure where each parameter has a clear interpretation (prox step size, strength of TV, balance between VCM and ICM, etc.).
>
> ---
>
> ## Weakness 3 – Dependence on the consistency / distillation strategy
>
> In a manner akin to LATINO (Spagnoletti et al., 2025), LVTINO is intentionally designed to leverage a consistency models (CMs) as prior distribution, but it is otherwise fully **agnostic to the specific training objective and distillation strategy** used to construct this model. For example, LVTINO can be deployed with CMs trained in isolation, derived from a diffusion model by distillation, as well as those distilled via adapters such as CM-LoRA. Similarly, it can be used with consistency-trajectory models that enhance conventional CMs via adversarial distillation. Moreover, given the close connections between LATINO and the concurrent Plug-and-Play Flow algorithm [^Martin], we expect that LVTINO could be directly extended to Flow Matching video priors at the expense of additional steps and hence a higher computational cost.
>
> [^Martin]: Martin, S. T., Gagneux, A., Hagemann, P., & Steidl, G. (2025). PnP-Flow: Plug-and-Play Image Restoration with Flow Matching. The Thirteenth International Conference on Learning Representations. Retrieved from https://openreview.net/forum?id=5AtHrq3B5R
>
> ---
>
> ## Weakness 4 – Trade-off between video and image regularizers
>
> The mixing hyperparameter the reviewer refers to controls the **relative influence between the VCM (video prior) and the ICM (image prior)** in the alternating scheme. We chose this for two reasons:
>
> 1. **Conceptual separation of roles.**
>
>    * The **VCM** enforces *temporal coherence* and global motion structure.
>    * The **ICM** focuses on *fine detail within frames* and visual perceptual quality.
>      Having a knob that interpolates between “more temporal” and “more spatial” information is natural and very interpretable.
>
> 2. **Complementary role of TV.**
>    The classical TV regularizer plays a different role: it acts as a **stabilizing, low-frequency smoother** that is particularly effective at removing small residual artifacts and noise. Empirically, once a reasonable TV weight has been chosen, varying it within a broad range has a smaller impact than changing the VCM/ICM balance.
>
> That said, we agree that the trade-off between **smoothing (TV) and image detail (ICM)** is also interesting. This is linked to the experiment shown in the plots above (see "Weakness 2" for the image link).
>
> ---
>
> We thank the reviewer again for the encouraging comments and helpful suggestions. We believe the planned changes will make the algorithm both clearer and more evidently robust.

---

> > ### Comment · Reviewer_acCb · 2025-11-25
> > **Thanks**
> >
> > Thanks for the rebuttal. I increase my score. It would be nice to point out the relation to [Martin et al, 2025] in a few more words. You would replace the prior steps using a flow matching denoising model? Maybe you can write how you can cast your framework into a PnP framework in the appendix, I would at least benefit from it.

---

> > > ### Author Response · Authors · 2025-11-29
> > >
> > > We thank the reviewer for informing us about the score increase, we truly appreciate it. We added Section A.5 to the Rebuttal PDF, linking our formulation to the PnP-Flow one introduced in [Martin et al., 2025].

---

### Official Review · Reviewer_WjZ3 · 2025-10-27

**Soundness:** 4
**Presentation:** 3
**Contribution:** 3
**Rating:** 6
**Confidence:** 5

**Summary:**

This work proposes a novel framework that enables high-definition video restoration in a plug-and-play manner using Video Consistency Models (VCMs).
Unlike previous methods that directly apply image LDMs to video inverse problems, the proposed approach explicitly captures temporal causality through VCMs. Leveraging the temporal priors encoded in VCMs, the method achieves substantial perceptual improvements over current state-of-the-art techniques.

Specifically, the video inverse problem formulated as $y=Ax+n$ is solved as follows.
The sampling process begins by initializing $x_0$ with the pseudo-inverse reconstruction $A^{\dagger}y$.
The subsequent diffusion sampling proceeds in three stages:

1. VCM sampling: the model first evolves the initial estimate using only the VCM prior for better temporal coherence.

2. Alternating VCM–ICM: the sampling then alternates between the VCM and Image Consistency Model (ICM) priors to progressively refine both temporal and spatial consistency.

3. ICM refinement phase: finally, the process is completed using the ICM prior alone for better spatial details.

Each after ICM sampling, a few iterations of the conjugate gradient (CG) method are performed to enforce data consistency,
while each after VCM sampling, the Adam optimizer is used to align the samples with the measurement constraint $y=Ax+n$.

**Strengths:**

1. Using Video Consistency Models (VCMs) to solve video inverse problems represents a novel and timely contribution, especially given the current interest on efficiency in high-definition video inverse problem solvers.

2. The idea of alternating between VCM and ICM is conceptually interesting, offering a fresh perspective on how temporal and spatial priors can be jointly leveraged.

3. The authors conduct comprehensive ablation studies, examining different design choices such as whether to enforce measurement consistency using the conjugate gradient or Adam optimizer, and whether to alternate between VCM and ICM priors or rely solely on VCM.
The results empirically demonstrate that alternating between VCM and ICM yields the most effective performance.

4. I really enjoyed reading the submission. The discussion of open problems provides valuable insights and motivates further research, making the paper timely and impactful.

**Weaknesses:**

1. I understand the empirical evidence showing that alternating VCM and ICM is effective for solving video inverse problems. However, there is no clear explanation as to why alternating between VCM and ICM yields better results than using VCM alone. Providing a clear theoretical or intuitive explanation would make the paper stronger.

2. The authors state that $p_\phi(x|\lambda)$, representing the total variation (TV) prior, helps prevent the ICM from introducing flickering artifacts. If that is the case, the TV regularizer should logically be applied after the ICM step. However, in the current formulation, $p_\phi(x|\lambda)$ is applied after the VCM step. Clarifying this design choice would be beneficial.

3. Since runtime and VRAM usage are also direct measures of a method’s efficiency, the authors should report these metrics for all comparative methods to ensure a fair and comprehensive comparison.

**Questions:**

1. Please provide the runtime and VRAM usage for all comparative methods to ensure a fair and comprehensive comparison.

2. Please provide a clear explanation addressing Weakness 1 and Weakness 2.

3. Please clarify whether the code will be publicly released, as this would enhance the reproducibility and impact of the work.

---

> ### Author Response · Authors · 2025-11-20
>
> We sincerely thank the reviewer for the careful and thorough assessment of our work, as well as for the many constructive comments and suggestions.
>
> ---
>
> ## Weakness 1 – Intuition for alternating VCM and ICM
>
> We believe that there are multiple explanations for why alternating ICM and VCM helps:
> - Using ICM alone makes problems involving temporal upsampling hard to tackle, as no notion of motion is known by the prior
> - Using VCM alone will for sure be the best choice for future works. Unluckily, current SOTA High-res distilled VCMs still suffer from lack of frame-wise details, and the results in our ablation study (Table 2) show this. With our work, we improve the restoration quality thanks to the ICM prior, and, to our knowledge, we are the first to propose a method that can synergically take advantage of both video-based and image-based priors.
>
> ---
>
> ## Weakness 2 – Order of the TV prior (TV after VCM vs. after ICM)
>
> This is a good point and is related to the way in which the mixture-of-experts is designed. Intuitively, putting the TV prior after the ICM will for sure reduce the flickering problem, but to do so a bigger lambda than the one used would be necessary. This will add the typical TV artifacts (flat areas in the frames), making thus ineffective the ICM efforts to add fine detail to the frames. Even if in this way we are propagating some flickering artifacts to the VCM step (that in any case are mitigated by the adding-removing noise process of the Stochastic Auto Encoder), the TV step after the VCM will take care of them. As here the proximal step and the VCM step have already helped in mitigating the flickering, it is possible to tune down the TV (a small lambda is enough), and avoid too strong TV artifacts.
>
> ---
>
> ## Weakness 3 – Runtime and VRAM usage
>
> We agree on this point. We will **add the following table** reporting **wall-clock time (seconds)** and **GPU memory usage (GB)** for each method, measured on the same hardware and for the same video: 25 frames at 1280x768 resolution.
>
> ### Runtime and memory usage
>
> | Method    | NFE↓ | Time per video (s)↓ | Peak GPU memory (GB)↓ |
> |-----------|-----:|--------------------:|----------------------:|
> | LVTINO    |  9   | 132                 | 35.15                 |
> | VISION-XL |  8   | 176                 | **15.64**             |
> | ADMM-TV   |  –   | **13.6**            | *22.01*               |
> | LVTINO-V  |  5   | *105*               | 25.42                 |
>
> ---
>
> ## Questions:
>
> ### Questions 1 and 2
> Addressed above (answers to weaknesses 1, 2, and 3)
>
> ### Question 3
> The code will be publicly released upon acceptance of this paper.
>
> ---
>
> We thank the reviewer again for the insightful comments and believe that the planned additions and clarifications will significantly improve the clarity, completeness, and impact of the paper.

---

### Official Review · Reviewer_Drrh · 2025-10-30

**Soundness:** 2
**Presentation:** 3
**Contribution:** 2
**Rating:** 4
**Confidence:** 3

**Summary:**

This paper proposes LVTINO, a zero-shot inverse solver for high-definition video restoration. It uses Video Consistency Models (VCMs) as a prior. The method combines a VCM prior (for temporal), an ICM prior (for spatial), and a classical TV regularizer in a unified framework. The method is based on a Langevin posterior sampler (like LATINO) and does not require gradient calculation through the network, making it efficient. The experiments show good results on several video restoration tasks like a combination of temporal super-resolution and deblurring.

**Strengths:**

1. LVTINO is computationally efficient. A very big advantage is that it does not need to compute gradients in the latent space, which is a heavy problem for methods like DPS (Chung et al., 2023). This is a very good point for practical use.

2. The paper combines the video prior (VCM), the image prior (ICM), and a classical prior (TV3) in a unified and clean way. I think this product-of-experts prior is an elegant and good way to solve this video restoration problem.

**Weaknesses:**

1. Lack of baseline comparison: This paper compares with only a very small number of baselines (VISION-XL, ADMM-TV). This is much less than other papers in this field. It is not necessary to solve all problems, but maybe the authors should focus on one task, for example Frame Interpolation (Temporal SR), and compare their method with more specialized models for that task. For example, comparing with BiM-VFI (Seo et al., in CVPR 25).

2. Limitied to *Linear* inverse problem: The method is proposed for linear inverse problems ($y = Ax + n$ where $A$ is a matrix). This is a fundamental limitation. It is not clear if it can be used for non-linear problems.

3. Somewhat impractical metric (NFE): The NFE (neural function evaluations) metric is not very practical (it is more like theoretical). For a better comparison, it would be good to add more practical metrics like runtime (seconds) and memory usage (GB) to Table 1.

4. Hyperparameter sensitivity: Table 2 shows that the optimal hyperparameters are very different for each problem (A, B, C). This suggests the method might be sensitive to these parameters. Also, the paper has not enough ablation study on how to choose these different hyperparameters (e.g., the lambdas for TV, eta, delta).

5. Missing Error Maps: The qualitative results (Figures 4, 5, 6) are good, but it would be much easier for readers to understand how well the model is performing if the authors provide error maps (e.g., Difference from GT).

**Questions:**

1. In Equation (2), the coefficient for the score term is $\beta_t$. But in Equation (3) (the ODE form), this coefficient changes to $\beta_t / 2$. Could you explain why there is a such change?

2. In Equation (4), the term $\tilde{x}_s$ is used in the integral, which is not defined (I can't find it). What is this variable?

3. On L175, the superscript (k) is used. Is superscript different from other subscripts k? Or is it just a typo?

4. On L213, the paper introduces a TV3 regularizer. Is this 3D Total Variation a widely used regularizer? It would be great if you can provide a reference for this.

---

> ### Author Response · Authors · 2025-11-20
>
> We thank the reviewer for the detailed feedback and for highlighting both the strengths and the limitations of our work.
> Below we address each weakness point-by-point and describe the additional experiments and analyses we will add to the paper.
>
> ---
>
> ## Weakness 1 – Lack of baseline comparison
>
> We agree that richer baselines are important. We thus extend the comparison as follows:
>
> - **New baseline: VIDUE.**
>   We add **VIDUE** as an additional baseline on the temporal blur + interpolation + SR×8 task (Problem C, as well as the corresponding Adobe/GoPro variants). VIDUE is one of the closest models to our setting, since it is explicitly trained for **joint motion-blur removal and frame interpolation**. This makes it a relevant reference point for LVTINO, which also solves a combined blur–interpolation problem. However, as it does not perform SR, we apply a x8 bicubic interpolation to upsample the outputs of VIDUE.
>
> - **Why we cannot include DiffIR2VR.**
>   A natural additional baseline would have been **DiffIR2VR**, which is the main competitor of VISION-XL in its original paper and uses **Stable Diffusion 2.1** as its prior. Unfortunately, the specific SD2.1 checkpoint relied upon by DiffIR2VR is no longer publicly available. This prevents us from reproducing their exact setup and therefore from including a fair, equal-to-equal comparison in our experiments.
>
> We will clearly state in the revised manuscript what mentioned here above.
>
> Concerning the suggested method, BiM-VFI (Seo et al., in CVPR 25), we noticed that it is a non-zero-shot method specifically trained for frame interpolation. This suggests an application of BiM-VFI to Problems B/C, where we also have a motion-blur and spatial SR component. To this extent we believe that comparing with VIDUE is more fair as it has been trained for joint motion-blur removal and frame interpolation, rather than frame interpolation only. In both cases a x8 bicubic interpolation to upsample the outputs is necessary. To prove this point we show here two frames obtained from the same observation $y$, resolved with the two models. We can obswerve that VIDUE removes some motion-blur while BiM-VFI does not.
>
> https://gcdnb.pbrd.co/images/L8XSnjgfkBwI.png?o=1
>
> ---
>
> ## Weakness 2 – Limited to linear inverse problems
>
> The formulation in the main text focuses on linear operators for clarity, but **LVTINO is not restricted to linear problems**:
>
> - In the **LATINO** paper (Spagnoletti et al., 2025), on which LVTINO builds, a closely related Langevin solver is tested on **non-linear forward operators**, and the Appendix E reports good performance in those settings.
> - The only requirement for LVTINO is that we can evaluate the (possibly approximate) **proximal operator** of the log-likelihood with respect to the unknown video. This is still possible in many non-linear cases.
>
> To make this point concrete, we add a new experiment:
>
> - **Noise + JPEG compression.**
>   We consider a degradation composed of additive Gaussian noise followed by **JPEG compression**, which is clearly non-linear and not representable as y = A x + n with a fixed matrix A. We apply LVTINO to this setting, keeping the same priors and modifying only the likelihood term to match the “noise + JPEG” forward model.
>
> In the revision, we will add a qualitative figure comparing:
>
> - **GT:** clean ground-truth video frames,
> - **y:** observations after Gaussian noise $\sigma=0.01$ + JPEG compression at level $=10$,
> - **LVTINO:** reconstructions obtained with our solver,
>
> including error maps.
>
> https://gcdnb.pbrd.co/images/5AcoxUsQJdf7.png?o=1
>
> Another example on the gopro240 dataset:
>
> https://gcdnb.pbrd.co/images/Ff1c2ziiFfq9.png?o=1
>
> This example, together with the results reported in LATINO’s appendix, demonstrates that the algorithm extends naturally to non-linear problems as long as the likelihood gradient can be approximated.
>
> ---
>
> ## Weakness 3 – Add runtime and memory usage
>
> We agree that **NFE alone is not sufficient** for practical assessment, even though it has the advantage of being hardware-agnostic. To address this:
>
> - We will **keep NFE** as a theoretical, model-agnostic measure of complexity.
> - We will **add table R4** reporting **wall-clock time (seconds)** and **GPU memory usage (GB)** for each method, measured on the same hardware and for the same problem size.
>
> ### Table R4 – Runtime and memory usage (to be added to the paper)
>
> | Method    | NFE↓ | Time per video (s)↓ | Peak GPU memory (GB)↓ |
> |-----------|-----:|--------------------:|----------------------:|
> | LVTINO    |  9   | 132                 | 35.15                 |
> | VISION-XL |  8   | 176                 | **15.64**             |
> | ADMM-TV   |  –   | **13.6**            | *22.01*               |
> | LVTINO-V  |  5   | *105*               | 25.42                 |

---

> ### Author Response · Authors · 2025-11-20
>
> ## Weakness 4 – Hyperparameter sensitivity
>
> We agree that hyperparameter robustness is an important practical concern. Our goal in Table 2 was to report **near-optimal** values per problem (averaged over the dataset), hence the differences. To better illustrate **stability**, we add the following analysis:
>
> - We focus on **Problem C**, which is the most challenging (temporal SR×8 + SR×8).
> - For a representative video, we vary **δ** (the step size) and **λ** around its default value and plot **PSNR** and **LPIPS**.
>
> In practice, we observe that:
>
> - Moderate changes in δ or λ only cause small variations in performance, indicating that LVTINO is **stable** and that **ε-good hyperparameters are easy to find** via coarse search. δ between $2e4$ and $2e5$ improves LPIPS and PSNR. The behavior w.r.t. λ is stable in the range considered, with a slight improvement in LPIPS around the value $1e-4$ (the one we use in our experiment).
>
> In the revised paper, we will include two plots:
>
> https://gcdnb.pbrd.co/images/UVzQ1A9QNgI7.png?o=1
>
> We will also clarify in the text that the hyperparameters reported in Table 2 were chosen after more extensive tests, and that the new plots illustrate a behavior supporting the claim that good hyperparameters can be found without exhaustive tuning.
>
> ---
>
> ## Weakness 5 – Error maps
>
> We agree that **error maps** are very informative, especially for visualizing where each method fails. In response, we will add:
>
> - **L2 error maps** (per-pixel squared error between reconstruction and ground truth) for a subset of the qualitative examples in Figures 4–6.
> - For each chosen sequence, we will show the GT, the reconstruction, and the corresponding error map for LVTINO, VISION-XL, and ADMM-TV.
>
> This will help highlight:
>
> - That LVTINO tends to reduce **structured temporal artifacts** and **residual blur**,
> - While VISION-XL and ADMM-TV exhibit larger errors around motion boundaries and fine structures.
>
> https://gcdnb.pbrd.co/images/gzZZy68G1Xba.png?o=1
>
> https://gcdnb.pbrd.co/images/KojQ5ZmCETPf.png?o=1
>
> https://gcdnb.pbrd.co/images/2dQkd7v6D8hG.png?o=1
>
> We hope that these additional experiments and clarifications address the reviewer’s concerns and make the strengths and practical relevance of LVTINO clearer.

---

> ### Author Response · Authors · 2025-11-20
>
> ## Questions:
>
> ### Question 1
> The change from equation (2) to equation (3) is correct. It is required for the ODE (3) and the SDE (2) to sample from the same probability distribution. See Song et al (2020).
>
> ### Question 2
> In equation (4), $x_s$ represents the solution of the Langevin diffusion process that converges to the posterior $p(x|y)$ as $s \rightarrow \infty$, given by the SDE
> $$ dx_s = \nabla \log p(x_s|y) + \sqrt{2} dw_s \quad \quad \text(A)$$
>
> which is the computational engine we use to produce posterior samples. Because this SDE cannot be solved in continuous time,  it is approximated by a discrete-time approximation $\{x_k\}_{k=1}^N$ that is constructed iteratively. The LATINO solver we use herein constructs this discrete-time approximation by using a splitting approach that involves the likelihood $p(y|x)$ and the prior $p(x)$ separately. In particular, while the likelihood $p(y|x)$ is involved via a traditional (implicit) gradient step or proximal step, the prior is involved via an auxiliary diffusion $\tilde{x}_s$ which solves a Langevin diffusion with the prior $p(x)$ as invariant distribution. Initialised at $\tilde{x}_0 = x_k$, this auxiliary diffusion is given by
>
> $$ d\tilde{x}_s = \nabla \log p(\tilde{x}_s) + \sqrt{2} dw_s \quad \quad \text(B)$$
> This will be clarified in the revised version of the manuscript.
>
> A key advantage of the LATINO strategy is that it allows embedding deep generative models via a stochastic autoencoding step, as opposed to conventional gradient strategies, which are suited to embedding score estimates or denoising operators via Tweedie's identity. In addition, the LATINO solver allows taking very long time steps and hence enjoys a faster convergence speed when compared to older Langevin samplers derived from a standard Euler-Maruyama discretization of $x_s$, such as the unadjusted Langevin algorithm (ULA), which require taking many small timesteps and hence converge more slowly. These two points will also be explained in more detail in the revised manuscript.
>
> ### Question 3
>
> Indeed, this is a typo that will be corrected in the final version.
>
> ### Question 4
>
> Spatio-temporal total variation (TV3) is a straightforward extension of the standard 2D total variation regularization [^ROF92]. It was introduced by Ng et al. (2007) [^Ng2007] as a refinement of the Tikhonov variant that was introduced by Shechtman [^Shechtman2005] as an effective tool to regularize the space-time super-resolution, motion deblurring, and temporal aliasing problems. This approach has been used in various ways in many subsequent papers [^Chan2011]. We shall add the reference to Ng et al (2007) in the final version of the manuscript.
>
> [^ROF92]: Rudin, Leonid I, Stanley Osher, and Emad Fatemi. 1992. “Nonlinear Total Variation Based Noise Removal Algorithms.” Physica D: Nonlinear Phenomena 60 (1–4): 259–68.
>
> [^Ng2007]: Ng, Michael K., Huanfeng Shen, Edmund Y. Lam, and Liangpei Zhang. 2007. “A Total Variation Regularization Based Super-Resolution Reconstruction Algorithm for Digital Video.” EURASIP Journal on Advances in Signal Processing 2007 (1): 074585. [doi:10.1155/2007/74585](https://doi.org/10.1155/2007/74585).
>
> [^Shechtman2005]: Shechtman, E., Y. Caspi, and M. Irani. 2005. “Space-Time Super-Resolution.” IEEE Transactions on Pattern Analysis and Machine Intelligence 27 (4): 531–45. [doi:10.1109/TPAMI.2005.85](https://doi.org/10.1109/TPAMI.2005.85).
>
> [^Chan2011]: Chan, S. H., R. Khoshabeh, K. B. Gibson, P. E. Gill, and T. Q. Nguyen. 2011. “An Augmented Lagrangian Method for Total Variation Video Restoration.” *IEEE Transactions on Image Processing* 20 (11): 3097–111. [doi:10.1109/TIP.2011.2158229](https://doi.org/10.1109/TIP.2011.2158229) [preprint](https://www.ccom.ucsd.edu/~peg/papers/ALvideopaper.pdf).

---

> > ### Comment · Reviewer_Drrh · 2025-11-23
> > **Problem w/ image link**
> >
> > Thanks for the rebuttal.
> >
> > However, provided links like https://gcdnb.pbrd.co/images do not work for me.
> > They just show `Bad Page`.
> >
> > Could you please find another way to deliver images or just indicate the images if they are already included in the revised pdf?
> >
> > Thank you.

---

> > > ### Author Response · Authors · 2025-11-23
> > >
> > > Thank you for the comment. We are sorry the content was not accessible to you. We used this web service because it preserves anonymity, and since OpenReview's markdown does not render images. We thought it would be accessible from anywhere in the world, but perhaps it is not, and we are sorry for that.
> > >
> > > We would be happy to upload the images to a web service of your choice, provided it ensures anonymity. Please let us know. You can also provide us with a link to a file drop service where we can upload the images, as long as it preserves both our and your identities.
> > >
> > > We will also update a revised pdf in the next days, but some images we shared in the comments are not intended to be added to the final manuscript (they are more to give a comparison/answer to a specific question).

---

> > > ### Author Response · Authors · 2025-11-25
> > >
> > > As a follow-up to our previous reply, we updated the rebuttal PDF, including new additions in red. We also added a new Appendix section (Section A.6), including the Images contained in the links (Figures 7, 8, 9, 10).

---

### Official Review · Reviewer_h2v5 · 2025-11-01

**Soundness:** 4
**Presentation:** 3
**Contribution:** 3
**Rating:** 6
**Confidence:** 3

**Summary:**

The paper proposed LATINO, a plug-and-play inverse solver for high definition video restoration with latent consistency model (LCM) priors. In particular, LATINO leveraged a video consistency model to capture global dependencies and causalities, and a image consistency model to recover local spatial detail and enhance perceptual quality on each frame. The two LCMs are integrated via a Langevin-type algorithm, together with measurement gradient guidance and regularizations. Experiments show superior performances comparing to baseline algorithms.

**Strengths:**

1. Successfully combined VCM and ICM to generate with both temporal causality and spatial quality.
2. Developed Langevin-type splitting algorithm to effectively balance measurement consistency with prior data consistency.
3. Demonstrated highly competitive experimental results under low NFEs.

**Weaknesses:**

1. The paper only shows experiments on a single dataset and limited baselines. Also, videos are short and noise level is low. Would be more convincing if more experiments could be added.
2. Since Langevin dynamics typically require the score function to be "well-behaved" to converge, fundamental questions could be raised on why LCM scores could lead to good sampling results. Further explanation should be provided to justify the choice of developing a Langevin-type algorithm for such tasks.
3. There seems to be no ablation studies on \eta, which is a crucial hyper-parameter in balancing the algorithm.
4. LATINO is based on various models and algorithms. It seems to me that introducing all base models dilutes the core motivation and method.

**Questions:**

1. Is it possible to experiment on more datasets and inverse problems?
2. Can the authors provide more insight of the Langevin dynamics algorithm?
3. Is it possible to carry out an ablation study on \eta?

---

> ### Author Response · Authors · 2025-11-20
>
> We thank the reviewer for the careful reading of our manuscript and the constructive comments.
> Below, we first summarize the new experiments and clarifications added during the rebuttal, and then answer each point in detail.
>
> * **Additional datasets / noise regimes.**
>   We now report results on (i) a *noisier* setting with Gaussian noise level $\sigma_n = 0.01$ on Adobe240, and (ii) a *new dataset*, GoPro240, for **all three inverse problems** A–C, using the same forward operators as in the main paper. These results are summarized in Tables R1–R2 and will be added to the main paper in Table 1 and/or in the appendix.
>
> * **New baseline (VIDUE).**
>   We include a comparison with **VIDUE**[^VIDUE] on the Temporal SR$\times 8$ + SR$\times 8$ task (Problem C), on both Adobe240 and GoPro240, which is a publicly available method that jointly addresses motion blur and temporal interpolation.
>
> * **Ablation on the mixing parameter $\eta$.**
>   We perform an ablation of the splitting weight $\eta$ that balances the VCM and ICM contributions in the Langevin scheme (Eq. (7)), see new Table R3.
>
> * **Clarifications to the text.**
>   We will add a short paragraph commenting on the role of $\eta$ and on the interpretation of the Langevin-type algorithm near Eqs. (6)–(7).
>
> [^VIDUE]: Shang *et al.* (2023), *VIDUE: Joint Video Multi-Frame Interpolation and Deblurring under Unknown Exposure Time*, **CVPR 2023**
> https://github.com/shangwei5/VIDUE
>
> ---
>
> ## Weakness 1 – “Single dataset and limited baselines”
>
> ### R1.1 Additional noise regime (σₙ = 0.01)
>
> The main paper focused on a mild noise regime σₙ = 0.001 to isolate the effect of the temporal/spatial inverse problem, and to put ourselves in a fair comparison with methods like Vision-XL which are conceived to work in the noiseless case only.
> To address the reviewer’s concern, we now include results with a **higher noise level σₙ = 0.01** on Adobe240 for **Problems A, B, and C**.
>
> Tables R1-A, R1-B and R1-C below mirror the layout of Table 1 and report FVMD, PSNR, SSIM and LPIPS for all methods and all three tasks:
>
> - LVTINO remains robust to increased noise, consistently outperforming ADMM-TV. Vision-XL, for the way it is designed does not allow for noisy measurments. Indeed, we observed NaN outputs when trying the original implementation on the considered inverse problems with additive Gaussian noise.
>
>
> ### R1.2 Additional dataset (GoPro240)
>
> We agree that testing on a second dataset is important. We therefore evaluate LVTINO, VISION-XL and ADMM-TV on 239 25-frames HR videos extracted from the **GoPro240** test dataset. We solve for **all three problems** A–C, using the same temporal SR and blur operators as on Adobe240.
>
> Tables R2-A, R2-B and R2-C show that:
>
> - LVTINO achieves the best FVMD and LPIPS across most tasks, indicating superior temporal consistency and perceptual quality.
> - On this dataset, which contains stronger real-world camera motion, LVTINO benefits substantially from the VCM prior, confirming that our approach generalizes.
>
> We hope that these new experiments address the concern about using only a single dataset and demonstrate that LVTINO generalizes well.
>
> ### R1.3 Discussion of additional baselines: DiffIR2VR vs VIDUE
>
> In the original VISION-XL paper, the main competing method is **DiffIR2VR**, which uses **Stable Diffusion 2.1** as its image prior. In our setting, faithfully reproducing DiffIR2VR is not currently possible as the specific SD2.1 checkpoint used as a prior is no longer directly available. As a result, we cannot rely on the same pretrained prior and therefore cannot include a fair DiffIR2VR baseline.
>
> Instead:
>
> - We introduce **VIDUE** as an additional baseline for the **Temporal SR$\times 8$ + SR$\times 8$ task (Problem C)** on both Adobe240 and GoPro240.
> - VIDUE is trained specifically for **joint motion-blur removal and temporal interpolation**, which makes it a natural comparator for our temporal blur + SR×8 operator.
> - However, **VIDUE is not trained for large-factor spatial super-resolution**, it is designed primarily for deblurring and frame interpolation, not for SR×8. Indeed, we perform a x8 bicubic interpolation to upsample the outputs of VIDUE. This explains why its PSNR/SSIM are lower than LVTINO, while still offering a good deblurring baseline.
>
> We will clarify this distinction in the paper to avoid any misunderstanding about (i) why a DiffIR2VR comparison is not feasible in our setup and (ii) why VIDUE is expected to underperform on the super-resolution component of the task. Furthermore, as stated in the article, the only other available zero-shot method for videos is **VDPS** [^Kwon2025], but it's unfeasible to deploy due to high memory requirements (>80Gb for 25 frames at 1280x768).
>
> [^Kwon2025]: Kwon et al., "Video Diffusion Posterior Sampling for Seeing Beyond Dynamic Scattering Layers," in IEEE Transactions on Pattern Analysis and Machine Intelligence, vol. 47.  [doi:10.1155/2007/74585].

---

> ### Author Response · Authors · 2025-11-20
>
> ## Tables for the new experiments
>
> ### Table R1-A – Adobe240, σₙ = 0.01, Problem A (Temporal SR×4 + SR×4)
>
> | Method   | NFE↓ | FVMD↓ | PSNR↑ | SSIM↑ | LPIPS↓ |
> |----------|-----:|------:|------:|------:|-------:|
> | LVTINO   | 9    | 256.5 | 24.95 | 0.782 | 0.331  |
> | VISION-XL| 8    | – | – | – | –  |
> | ADMM-TV  | –    | 424.1 | 17.85 | 0.758 | 0.373  |
>
>
> ---
>
> ### Table R1-B – Adobe240, σₙ = 0.01, Problem B (Temporal blur + SR×8)
>
> | Method   | NFE↓ | FVMD↓ | PSNR↑ | SSIM↑ | LPIPS↓ |
> |----------|-----:|------:|------:|------:|-------:|
> | LVTINO   | 9    | 62.6 | 21.91 | 0.671 | 0.448  |
> | VISION-XL| 8    | – | – | – | –  |
> | ADMM-TV  | –    | 145.5 | 21.35 | 0.646 | 0.471  |
>
> ---
>
> ### Table R1-C – Adobe240, σₙ = 0.01, Problem C (Temporal SR×8 + SR×8)
>
> | Method   | NFE↓ | FVMD↓ | PSNR↑ | SSIM↑ | LPIPS↓ |
> |----------|-----:|------:|------:|------:|-------:|
> | LVTINO   | 7    | 310.6 | 23.42 | 0.688 | 0.428  |
> | VISION-XL| 8    | – | – | – | –  |
> | VIDUE    | 1    | 121.5 | 21.33 | 0.603 | 0.511  |
> | ADMM-TV  | –    | 1665 | 18.12 | 0.652 | 0.475  |
>
> ---
>
> ### Table R2-A – GoPro240, Problem A (Temporal SR×4 + SR×4)
>
> | Method    | NFE↓ | FVMD↓ | PSNR↑ | SSIM↑ | LPIPS↓ |
> | --------- | ----:| -----:| -----:| -----:| ------:|
> | LVTINO    |    9 | 189.4 | 24.01 | 0.775 |  0.315 |
> | VISION-XL |    8 | 282.2 | 26.06 | 0.792 |  0.326 |
> | ADMM-TV   |    – | 265.8 | 24.32 | 0.745 |  0.406 |
>
>
> ---
>
> ### Table R2-B – GoPro240, Problem B (Temporal blur + SR×8)
>
> | Method   | NFE↓ | FVMD↓ | PSNR↑ | SSIM↑ | LPIPS↓ |
> |----------|-----:|------:|------:|------:|-------:|
> | LVTINO   | 9    | 46.20 | 22.46 | 0.687 | 0.433  |
> | VISION-XL| 8    | 52.03 | 24.05 | 0.697 | 0.486  |
> | ADMM-TV  | –    | 145.9 | 20.83 | 0.618 | 0.527  |
>
> ---
>
> ### Table R2-C – GoPro240, Problem C (Temporal SR×8 + SR×8)
>
> | Method   | NFE↓ | FVMD↓ | PSNR↑ | SSIM↑ | LPIPS↓ |
> |----------|-----:|------:|------:|------:|-------:|
> | LVTINO   | 7    | 232.6 | 22.91 | 0.677 | 0.445  |
> | VISION-XL| 8    | 995.9 | 22.67 | 0.669 | 0.474  |
> | VIDUE    | 1    | 84.45 | 20.66 | 0.571 | 0.548  |
> | ADMM-TV  | –    | 969.3 | 17.70 | 0.631 | 0.527  |

---

> ### Author Response · Authors · 2025-11-20
>
> ## Weakness 2 – “Langevin dynamics … well-behaved scores”
>
> The theory of the Langevin SDE underpinning LVTINO does indeed require the likelihood gradients and prior scores to be well defined and Lipschitz continuous over the latent space. In the case where only one prior is used, either the video CM or the per-frame image CM, then one can rewrite the SDE directly in the latent space of that model and examine the regularity properties of the latent CM network and of the (latent) data-fidelity operator defined by concatenating the decoder-prox-encoder steps. Under some technical assumptions, the required regularity assumption on the latent CM network will follow from the regularity of the PF-ODE and the CM architecture used, whereas the regularity of the decoder-prox-encoder steps depends predominantly on the choice of the architecture used for the auto-encoder. However, LVTINO requires theory that simultaneously considers both the video CM and the per-frame image CM, when these models do not share the same latent space. This is significantly more challenging as it requires operating with degenerate diffusions (see [^Cass]). As a result, the theoretical analysis of LVTINO is exceeding technical and has been postponed to future work. We will clarify this important point in the revised manuscript.
>
> More generally, we are aware of an ongoing effort to study the theoretical and empirical convergence properties of LATINO as a Langevin sampler in the long-time regime (e.g., ergodicity, convergence to stationarity, asymptotic bias, etc). Some empirical results where recently reported in Appendix D of https://arxiv.org/pdf/2507.02686, where a simplified LATINO sampler was run for 50,000 iterations and shown to be numerically stable and ergodic. This analysis is also relevant to LVTINO when only one prior is used, either the video CM or the per-frame image CM, but it does not yet cover the mixture-of-experts case. We hope and anticipate that our manuscript and the strong empirical performance of LVTINO as a few-step sampler will stimulate research into its theoretical properties, including convergence guarantees.
>
> [^Cass]: Thomas Cass. Dan Crisan. Paul Dobson. Michela Ottobre. "Long-time behaviour of degenerate diffusions: UFG-type SDEs and time-inhomogeneous hypoelliptic processes." Electron. J. Probab. 26 1 - 72, 2021. https://doi.org/10.1214/20-EJP577
>
> ---
>
> ## Weakness 3 – “No ablation on η”
>
> The hyperparameter $\eta$, which controls the balance between the VCM and the ICM in our theoretical framework, must be translated in practice through the choice of the corresponding evaluation times $t_V$ and $t_I$. In particular, when $\eta$ increases the $t_V$ is larger and the ICM is evaluated at smaller $t_I$. Because pretrained Consistency Models are only accurate on a restricted subset of timesteps, this severely limits how finely we can tune $\eta$ in practice.
>
> To approximate different effective values of $\eta$, we therefore perform an ablation where we vary the possible video timesteps $t_V$ and image timesteps $t_I$ that remain within the valid finetuned ranges of the two backbones. Operationally, we choose among the subsets:
>
> * **VCM (video model):** ([757, 522, 375, 255, 125])
> * **ICM (image model):** ([749, 624, 499, 374, 249, 124, 63])
>
> and pairing them to simulate “larger $\eta$” (larger VCM steps + smaller ICM steps) and “smaller $\eta$” (smaller VCM steps + larger ICM steps).
>
> ### Table R3: **Experimental configurations tested**
>
> For clarity, here we show some configurations evaluated to get a comparison on Problem B (Temporal SR$\times 4$ + SR$\times 4$). Each configuration is evaluated on the same sample sequence:
>
> | Experiment   | $t_V$ (video)             | $t_I$ (image)  | PSNR ↑    | SSIM ↑   | LPIPS ↓  |
> | ------------ | ------------------------- | ------------------- | --------- | -------- | -------- |
> | **EXP 1**    | [375, 255, 125]           | [499, 374]          | 22.90 | 0.752 | 0.296 |
> | **EXP 2**    | [757, 522, 375]           | [249, 124]          | 22.56 | 0.714 | 0.308 |
> | **EXP 3**    | [522, 375, 255]           | [374, 249]          | 22.80 | 0.762 | 0.290 |
> | **EXP 4**    | [522, 255, 125]           | [749, 624]          | 22.36 | 0.720 | 0.317 |
> | **BASELINE** | [757, 522, 375, 255, 125] | [374, 249, 124, 63] | 23.96 | 0.770 | 0.272 |
> | **VISION-XL**  |          ---            |          ---        | 24.36 | 0.667 | 0.488 |
>
> For comparison, the values used for the experiments shown in the article are: $t_V\in$ [757, 522, 375, 255, 125] and $t_I \in$ [374, 249, 124, 63]. We notice how, even with fewer steps and varying the configurations, the metrics remain stable.

---

> ### Author Response · Authors · 2025-11-20
>
> ## Weakness 4 – “Based on various models and algorithms … core motivation diluted”
>
> Our goal is to introduce a **generic plug-and-play inverse solver** able to exploit *any* off-the-shelf CMs as priors. In our experiments, we instantiate:
>
> - A single VCM (CausVid) for temporal dynamics,
> - A single ICM (DMD2) for spatial detail, and
> - A simple convex TV³ regularizer for stability.
>
> We will emphasize in the introduction and method section that LVTINO is a Langevin-based inverse solver whose structure is independent of the specific CM backbones. The ablations in the paper and the new η ablation show that each component plays a specific role, and that the VCM-only variant remains a competitive lighter model when only temporal coherence is prioritized. We hope that future work can further exploit our formulation by adapting each component to specific tasks, as the Langevin formulation we adopted, together with the reduced number of steps required, makes it well-suited for fine-tuning the different components.

---

### Author Response · Authors · 2025-11-29
**Summary of the rebuttal process**

We would like to thank all the reviewers for their efforts in highlighting the strengths of our work and for raising very interesting questions that helped us improve the soundness of our presentation.

Even if the process was unfortunately cut short beforehand by external events, we hope our replies provided the requested clarifications satisfactorily.

To help the assigned Area Chair with his work, we provide a brief summary of the additions/modifications adopted in the current rebuttal version of the paper and how they relate to the reviewers' requests.

The $\color{red}\text{red}$ lines and sections are the additions compared to the submitted version of the paper. In particular, we highlight:

* **in the main paper:** In section 4, we updated the **Datasets & Metrics** paragraph to take into account the newly added tests on another dataset (the GoPRO240 one, with the results shown in Table 2), as requested by Reviewer h2v5. In a similar way, we updated Table 1 with a new baseline for Problem C (VIDUE) as requested by both Reviewer h2v5 and Reviewer Drrh. We also added Table 3 and a **Computational Efficiency** paragraph, as requested by both Reviewer Drrh and Reviewer WjZ3.
* **the newly added Section A.5** that highlights the connection between PnP-Flow[^Martin] and LATINO[^Spagnoletti], the method on which our work is based. In particular, we explain how our approach can be adapted to the Flow Matching framework. This addition has been requested by Reviewer acCb during our discussion.
* **the newly added Section A.7** with additional experiments. As requested by Reviewer Drrh, we added experiments on noisier scenarios. While our setting allows for handling noisy cases, the current SOTA baseline, VISION-XL, does not; for this reason, we limited ourselves to a mild noise case. We show in Table 6 that even with more aggressive noise, we still beat on all the metrics classical baselines like ADMM-TV or the VIDUE model for Problem C. We also add a more extensive discussion in the paragraph **Comparisons to other baselines** on why no other baseline comparisons are possible and why we preferred the VIDUE baseline to the BiM-VFI one proposed by Reviewer Drrh. As another concern raised by Reviewer Drrh was linked to the non-linear operators case, we show how this is not a limitation, as our model can successfully solve a noise + JPEG compression problem, which is non-linear. In the paragraph **Hyperparameter Sensitivity**, we address the concerns of Reviewers h2v5 and Drrh on the difficulty of aligning the hyperparameters introduced in our formulation. We show in Table 7 and Figure 9 that, even with drastically different hyperparameters $\delta, \lambda$ and $\eta$, the performance remains stable, and that with a simple grid search it is possible to obtain $\epsilon$-optimal results. Finally, as suggested by Reviewer Drrh, we added $ L_2$ error plots to the **Error Map Analysis** paragraph to further demonstrate the improvements in reconstruction quality achieved by our method.

We are aware that the anonymous links to the images used in our replies may not be accessible everywhere on earth, so we included those images in Section A.7.

We finally state that we plan to carefully assess minor changes highlighted by the reviewers and improve the clarity of the exposition to improve readability. Upon acceptance, the code to reproduce the results will be publicly released.

[^Martin]: Martin, S. T., Gagneux, A., Hagemann, P., & Steidl, G. (2025). PnP-Flow: Plug-and-Play Image Restoration with Flow Matching. The Thirteenth International Conference on Learning Representations. Retrieved from https://openreview.net/forum?id=5AtHrq3B5R

[^Spagnoletti]: Spagnoletti, A., Prost, J., Almansa, A., Papadakis, N., & Pereyra, M. (2025). LATINO-PRO: LAtent consisTency INverse sOlver with PRompt Optimization. Proceedings of the IEEE/CVF International Conference on Computer Vision (ICCV).

---

### Meta-Review · Area_Chair_QWcH · 2026-01-07

**Summary:**

Reviewers mostly agree the paper is timely and novel: it proposes a zero-shot / plug-and-play framework for high-definition video restoration that leverages Video Consistency Models (VCMs) to capture temporal causality and couples them with an Image Consistency Model (ICM) plus a classical TV³ prior. The approach is praised for being efficient and practical, notably avoiding backpropagation through the generator while achieving strong perceptual and temporal quality with few steps.

The main concerns originally focused on evaluation breadth (datasets, baselines, and settings), practical metrics (runtime and VRAM vs NFE), hyperparameter sensitivity, and method clarity and justification. The rebuttal substantially addressed these concerns through additional experiments, expanded efficiency reporting, and targeted ablation studies, with one reviewer (acCb) explicitly increasing their score. While some issues remain regarding baseline completeness and presentation clarity, the strengths and practical impact of the proposed method outweigh these residual concerns. The AC therefore recommends acceptance and expects the authors to address the remaining issues in the final revision.

**Reviewer Concerns:**

### Addressed

**Practical efficiency metrics:** Added runtime and VRAM reporting, still show superiority (WjZ3, Drrh).

**Ablations / sensitivity:** Added studies targeting key parameters and stability (h2v5, Drrh, acCb).

**Qualitative diagnostics:** Added error maps and more evidence to help readers identify where improvements come from (Drrh).

**Clarity improvements (committed / partially delivered)**: Planned restructuring to reduce density and move derivations to the appendix; acCb was satisfied enough to raise the score.

========================================

### Still outstanding

**Experiments completeness (h2v5, Drrh, WjZ3):** a) Even with added baselines, comparisons remain narrow. AC suggests that the authors at least discuss the literature. b) For the nonlinear degradation demonstration, the rebuttal provides evidence but lacks a direct comparison to competitive methods tailored to that setting. c) Runtime/efficiency should be contextualized against the most relevant available methods (e.g., VIDUE, and other zero-shot/video approaches). d) Results of more settings, like long videos, and evaluation on additional datasets and more challenging settings are required.

**Succinctness (h2v5, acCb)**: The framework’s reliance on multiple pretrained models (VCM + ICM + TV³  + solver choices) can still read as “assembled,” potentially diluting the central conceptual message unless the paper sharply emphasizes what is novel and what is an instantiation choice.

**Deeper rationale for design choices (WjZ3)**: The rebuttal provides intuition, but the paper would still benefit from a clearer explanation with evidence for why alternating VCM and ICM improves over VCM-only, and why the TV prior is applied in the chosen order, etc.

**Reviewer Scores:**

Reviewer h2v5 (6 $\rightarrow$ 6): Initially concerned about limited datasets/baselines, simplified settings, and missing parameter studies. The rebuttal addresses most issues with added experiments and ablations, but the concern that relying on multiple base models dilutes the core methodological message likely remains. Besides, there is no discussion/results on long videos.

Reviewer Drrh (4 $\rightarrow$ 4/6): Raised issues about narrow baselines, linear degradation assumptions, lack of practical metrics, and missing diagnostics (error map). The rebuttal substantially improves these points, but the nonlinear degradation results still lack comparison with competitive methods.

Reviewer WjZ3 (6 $\rightarrow$ 6/8): Praises the novelty of using VCMs for HD video inverse problems and finds the VCM/ICM alternation and additional discussion insightful. Most design and efficiency concerns are addressed in the rebuttal, though more runtime comparisons with relevant baselines (e.g., VIDUE, DiffIR2VR where feasible) would further strengthen the case.

Reviewer acCb (6 $\rightarrow$ 8): Strongly appreciated the novelty and lightweight, gradient-free design. Concerns about robustness and clarity were addressed in the rebuttal, leading the reviewer to explicitly increase the score.

---

### Decision · Program_Chairs · 2026-01-26

Accept (Poster)